# Pan-Arctic surface ozone: modelling vs measurements

Xin Yang[1], Anne-M. Blechschmidt[2], Kristof Bognar[3], Audra McClure–Begley[4,5], Sara Morris[4,5], Irina Petropavlovskikh[4,5], Andreas Richter[2], Henrik Skov[6], Kimberly Strong[3], David W. Tarasick[7], Taneil Uttal[5], Mika Vestenius[8], Xiaoyi Zhao[7]

[1]British Antarctic Survey, UK Research Innovation, Cambridge, UK
[2]Institute of Environmental Physics, University of Bremen, Bremen, Germany
[3]Department of Physics, University of Toronto, Toronto, ON, Canada
[4]Cooperative Institute for Research in Environmental Sciences, University of Colorado, Boulder, CO, USA
[5]NOAA Earth System Research Laboratory, Boulder, CO, USA
[6] iClimate, Department of Environmental Science, Aarhus University, Denmark
[7]Air Quality Research Division, Environment and Climate Change Canada, Toronto, ON, Canada
[8]Atmopsheric Composition Research, Finnish Meteorological Institute, Helsinki, Finland

*Correspondence to*: Xin Yang (xinyang55@bas.ac.uk)

**Abstract.** Within the framework of the International Arctic Systems for Observing the Atmosphere (IASOA), we report a modelling-based study on surface ozone across the Arctic. We use surface ozone from six sites: Summit (Greenland), Pallas (Finland), Barrow (USA), Alert (Canada), Tiksi (Russia), and Villum Research Station (VRS) at Station Nord (North Greenland, Danish Realm), and ozonesonde data from three Canadian sites: Resolute, Eureka, and Alert. Two global chemistry models: a global chemistry transport model (p-TOMCAT) and a global chemistry climate model (UKCA), are used for model-data comparisons. Remotely sensed data of BrO from the GOME-2 satellite instrument and ground-based Multi-axis Differential Optical Absorption Spectroscopy (MAX-DOAS) at Eureka, Canada are used for model validation.

The observed climatology data show that spring surface ozone at coastal sites is heavily depleted, making ozone seasonality at Arctic coastal sites distinctly different from that at inland sites. Model simulations show that surface ozone can be greatly reduced by bromine chemistry. In April, bromine chemistry can cause a net ozone loss (monthly mean) of 10-20 ppbv, with almost half attributable to open-ocean-sourced bromine and the rest to sea-ice-sourced bromine. However, the open-ocean-sourced bromine, via sea spray bromide depletion, cannot by itself produce ozone depletion events (ODEs) (defined as ozone volume mixing ratios VMRs < 10 ppbv). In contrast, sea-ice-sourced bromine, via sea salt aerosol (SSA) production from blowing snow, can produce ODEs even without bromine from sea spray, highlighting the importance of sea ice surface in polar boundary layer chemistry.

Modelled total inorganic bromine (Br$_Y$) over the Arctic sea ice is sensitive to model configuration, e.g., under the same bromine loading, Br$_Y$ in the Arctic spring boundary layer in the p-TOMCAT control run (i.e., with all bromine emissions) can be 2 times that in the UKCA control run. Despite the model differences, both model control runs can successfully reproduce large bromine explosion events (BEEs) and ODEs in polar spring. Model-integrated tropospheric column BrO generally matches GOME-2 tropospheric columns within ~50% in UKCA and a factor of 2 in p-TOMCAT. The success of the models in reproducing both ODEs and BEEs in the Arctic indicates that the relevant parameterizations implemented in the models work reasonably well, which supports the proposed mechanism of SSA production and bromide release on sea ice. Given that sea ice is a large source of SSA and halogens, changes in sea ice type and extent in a warming climate will influence Arctic boundary layer chemistry, including the oxidation of atmospheric elemental mercury. Note that this work dose not necessary rule out other possibilities that may act as a source of reactive bromine from sea ice zone.

## 1 Introduction

Climatological data show that mean surface ozone across the Arctic is ~5 ppbv higher than that in the Antarctic (Helmig et al., 2007), reflecting the impact of anthropogenic emissions of ozone precursors such as NOx (=NO+NO$_2$) and volatile organic compounds (VOC) in the northern hemisphere (NH) (e.g., Law and Stohl, 2007; Quinn et al., 2008; Walker et al., 2012; Ancellet et al., 2016). For a specific location, the surface ozone depends on multiple factors, including the elevation above sea level (asl), proximity to the coast, human influence, and processes such as photochemical production/loss rates and dry deposition. Over the past several decades, Arctic sea ice extent has been declining (e.g., Cavalier et al., 2012; Laxon et al., 2013, Olonscheck et al., 2019) and thinning (Lindsay and Zhang 2005; Kwok and Rothrock, 2009). The rapid disappearance of summer multiyear sea ice means there will be more young sea ice in the following winter and spring, which will potentially affect the exchange of chemical compounds (both gaseous and particulate phase) between the ocean, sea ice, and the atmosphere. A modelling study shows that the alteration of surface albedo alone, in a scenario of a sea-ice-free Arctic summer, can significantly alter the atmospheric oxidizing capacity at high latitudes, including the concentrations of ozone and the hydroxyl radical (OH) (Voulgarakis et al., 2009a). Therefore, the rapid change of Arctic environment in a warming climate may greatly affect Arctic near surface ozone concentration, seasonality and long-term trend (Tarasick and Bottenheim 2002).

Observations of anomalously low boundary layer ozone at coastal sites in the Arctic spring have been reported (Bottenheim et al., 1986; Barrie et al., 1988). An ozone depletion event (ODE) often refers to surface ozone volume mixing ratios (VMRs) drops < 10 ppbv or even near zero levels. ODEs are mostly found in association with strongly enhanced bromine, so-called bromine explosion events (BEEs). The enhanced bromine monoxide (BrO) can extend from near the surface to a height of a few km, as has been frequently observed by in situ measurements (e.g., Liao et al., 2011, 2012; Buys et al., 2013; Schultz et al., 2017 and references therein), ground-based remote sensing (e.g., MAX-DOAS, Frieβ et al, 2011; Zhao et al., 2016) and satellite-based remote sensors (e.g., Wagner and Platt, 1998; Theys et al., 2011). Analyses of Arctic transport (Bottenheim and Chan 2006; Liu et al., 2013), as well as in situ measurements (Bottenheim et al., 2009; Jacobi et al., 2010; Seabrook et al., 2013) suggest that the near-surface ozone minimum in spring is not limited to coastal sites, but covers much of the Arctic basin, indicating that the sources of bromine are mainly sea-ice-related (e.g., Simpson et al. 2007a, Abbatt et al., 2012). However, the dominant sources of bromine during ODEs or BEEs are still under debate. Proposed candidates for reactive bromine release include e.g., frost flowers (Kaleschke et al. 2004), first-year sea ice (Skov et al., 2004; Simpson et al., 2007b), sea salt aerosol (SSA) produced from blowing snow (Yang et al., 2008), snowpack (Pratt et al., 2013; Custard et al., 2017), SSA from open leads (e.g., Kirpes et al., 2019). For example, Pratt et al. (2013) showed that the snowpack is a source of reactive halogens. Custard et al. (2017) provided further evidence that snowpack activation occurs. In addition, stratospheric BrO intrusions in association with downward transport of air masses from lower stratosphere also affect polar free tropospheric BrO (Salawitch et al., 2010). Global chemical models have been used to test chemical schemes for interpreting or reproducing observed spring ODEs and BEEs. For instance, Toyota et al. (2011) and Falk and Sinnhuber (2018) focused on snowpack-released bromine while Yang et al. (2010) and Choi et al. (2012; 2018) considered blowing-snow-sourced bromine. Box (or 0-D) models are used for process studies such as heterogeneous reactions on various saline particles including SSA, frost flowers, and snowpack (Fan and Jacob 1992; Tang and McConnell, 1996; Michalowski et al. 2000; Evans et al., 2003). 1-D models have also been developed with a focus on the exchange of gaseous-phase halogens between the air in the boundary layer and the snowpack, and boundary layer ODEs and BEEs (Saiz-Lopez et al., 2008; Thomas et al., 2011, 2012; Cao et al. 2013; 2016).

Recent winter cruise data from the Weddell Sea, Antarctica, confirm that the sea ice surface is a large source of sea salt aerosol (Frey et al., 2019; Yang et al., 2019). Like the open-ocean-sourced sea spray, the sea-ice-sourced SSA is also a large reservoir of various chemical compounds, including inorganic halogens. Through heterogeneous reactions, bromide (Br⁻) and chloride (Cl⁻) can be activated and released to the air to form a large source of inorganic halogens (Fan and Jacob, 1992, Vogt et al.,

1996), and the consequences may induce polar boundary layer bromine explosion events and ozone depletion events (Simpson et al., 2007a; Abbatt et al., 2012).

SSA bromide data collected in the NH mid-to-low latitudes show that bromide is largely depleted with respect to sodium without a clear seasonal cycle of the depletion strength (Sander et al., 2003). This is attributed to the air pollution and

5 acidification of SSA in the NH. However, in the Southern Ocean of Antarctica, where the air is less polluted, a seasonally varying bromide depletion strength is observed (Ayers et al., 1999, Legrand et al., 2016) with maximum depletion factors in later spring to early summer and a minimum in winter. Global chemistry models with a detailed tropospheric bromine scheme show that open-ocean-sourced bromine can cause tropospheric ozone loss of ~5% at mid- to-low latitudes and up to 15-30% at high latitudes (Yang et al., 2005; Parrella et al., 2012). Model runs with sea-ice-sourced bromine implemented show that

an additional 10-25% ozone loss can be simulated in polar spring (Yang et al., 2010). Global models with a relatively coarse horizontal resolution of a few degrees by a few degrees can explain large-scale (e.g., > ~500 km) ODEs and BEEs in both polar regions (Theys et al., 2011; Zhao et al., 2016; Legrand et al. 2016; Choi et al., 2018). However, no systematic validation against measured ozone and BrO in the Arctic and the Antarctic has been presented. This is important to examine and refine the bromine scheme implemented in models, especially in the polar regions.

Most current global-scale chemistry models do not have sea-ice-sourced halogens included. A recent assessment of tropospheric ozone performance in current global models is mainly focused on mid-latitudes (Young et al., 2018), as are most global ozone seasonality studies (e.g., Derwent et al., 2016; Parrish et al., 2016). Previous multi-model assessments of Arctic surface ozone in global chemistry transport models (CTMs) gave quite different implications on the role of halogens. For instance, Monks et al. (2015) and early modelling work by Shindell et al. (2008) showed over-prediction of surface ozone at

Barrow in spring, implying a result of missing halogen chemistry. However, Emmons et al. (2015) showed a general model under-prediction in April compared with ozone sondes, suggesting that the halogen-induced bias may not be pervasive in the Arctic troposphere. In a very recent modelling work focusing on polar tropospheric halogens (Fernandez et al., 2019), photochemical release of molecular bromine, chlorine, and interhalogens from the sea-ice surface, and iodine biologically produced underneath and within porous sea-ice are considered. However, relatively little is known about model skill in

reproducing polar spring boundary layer ozone, especially on short-time scales of hourly, daily, leaving a large gap in our understanding of the global ozone budget in the polar regions.

Although observations of surface ozone and tropospheric vertical ozone profiles are limited in the Arctic, existing data clearly show that there is a spring ozone maximum at inland sites such as Pallas, Finland (e.g., Hatakka et al., 2003) and Summit, Greenland (3208 m asl) (e.g., Helmig et al., 2007). It has been proposed that this spring ozone maximum, also seen at other

high-latitude locations (e.g., Monks et al., 2000), is attributable to reduced ozone photo-dissociation and dry deposition in winter, balanced by increased stratospheric ozone intrusions in spring following the breakup of the polar vortex in the lower stratosphere (e.g., Laurila 1999; Helmig et al., 2007). However, at coastal sites, ozone is observed to be heavily depleted during spring. Moreover, the near-surface ozone minimum observed in spring is not limited to coastal sites, but covers much of the Arctic boundary layer (Liu et al., 2013; Hardacre et al., 2014). Can global models with state-of-the-art bromine chemistry

reproduce this pan-Arctic spring ozone depletion? What is the dominant factor that causes spring ODEs and BEEs? These are the two key questions addressed in this study.

We employ multi-year integrations in two global chemistry models (the p-TOMCAT chemistry transport model and the UKCA chemistry-climate model) and perform comparisons to observations of surface ozone, vertical ozone profiles, and GOME-2 tropospheric column BrO, in order to validate the effect of these modelled processes on ozone depletion and BrO enhancement.

This work is undertaken in the framework of International Arctic Systems for Observing the Atmosphere (IASOA), whose mission is to advance coordinated and collaborative research objectives using data from independent Arctic atmospheric observatories (Uttal et al. 2016). This is a modelling-based study of the pan-Arctic surface zone; more information about ozone climatology can be found in a companion paper by McClure–Begley et al. (in preparation). The surface ozone climatology

data used in this study are from Summit, Greenland (72.6°N, 38.5°W), Pallas, Finland (68.0°N, 24.1°E), Barrow, USA (71.3°N, 156.6°W), Alert, Canada (82.5°N, 62.3°W), Tiksi, Russia (71.6°N, 128.9°E) and Villum Research Station (VRS), at Station Nord, Greenland (81.4°N, 16.4°W). Ozonesonde data are from three Canadian sites: Resolute (74.7°N, 95.0°W), Eureka (80.1°N, 86.4°W), and Alert. Retrievals of tropospheric column BrO from the GOME-2 instrument, including maps and subsetted data for each site, and ground-based MAX-DOAS BrO at Eureka are also used. Figure 1 shows the locations of these sites. Measurements are described briefly in Section 2. Model experiments are described in Section 3. The results of the model-data comparison is presented in Sections 4. Discussions and summary are in section 5 and 6, respectively.

## 2 Measurements

### 2.1 Surface ozone and ozonesondes

Surface ozone data are retrieved from the World Data Centre for Reactive Gases (WDCRG) and archived at the NOAA Global Monitoring Laboratory (https://www.esrl.noaa.gov/gmd/ozwv/surfoz/data.html). The measurements of surface ozone are made by several brands of dual cell UV absorption monitors, which relate UV absorption to ozone concentration following the Beer-Lambert law. Details can be found in these articles: e.g., VRS Research Station in Skov et al. (2004, 2019), Alert in Bottenheim et al. (2002), or in review articles by e.g., Gaudel et al. (2018), Oltmans et al. (2010) and Cooper et al. (submitted). In general, the technique has a detection limit of about 1 ppbv and an uncertainty (95% confidence interval) of about 1 ppbv for VMRs below 10 ppbv and about 2 ppbv for more typical surface VMRs of 30-40 ppbv (Galbally et al., 2013; Tarasick et al., 2019a).

Ozonesonde data from the three Canadian stations used here can be found at the World Ozone and Ultraviolet Radiation Data Centre (WOUDC). During the period of interest here, all ozonesondes used were electrochemical concentration cells (ECC) (Komhyr, 1969), manufactured by Environmental Science (EN-SCI) Corp. All sondes used the conventional neutral-buffered 1% potassium iodide sensing solution. The data records of the Canadian sites have recently been re-evaluated (Tarasick et al., 2016). Based on the typical ozone sensor response time of 25-40 s (Smit and Kley, 1998), and assuming a typical balloon ascent rate of 4-5 m s$^{-1}$, the ozonesondes have a vertical resolution of about 100-200 metres. Measurement precision is ±3-5% and the overall uncertainty in ozone VMRs is less than 10% in the troposphere (Kerr et al., 1994; Smit et al., 2007; Tarasick et al., 2016; 2019a, c).

Ozonesonde releases are normally once per week, although additional releases are often scheduled during observational campaigns in the Arctic spring. Despite their low frequency of observation compared to surface monitoring, ozonesondes have been used successfully to study boundary-layer processes like ODEs (e.g., Bottenheim et al., 2002; Tarasick and Bottenheim, 2002) and long-range transport (e.g., Oltmans et al., 2010; He et al., 2011; Tarasick et al., 2019b).

### 2.2 Complementary data sets

In addition to the ozone measurements, several other data sets are employed in this study: tropospheric columns of BrO from the Global Ozone Monitoring Experiment-2 (GOME-2; Callies et al., 2000) instrument onboard the Meteorological Operational Satellite-A (MetOp-A), and lower tropospheric profiles of BrO from ground-based Multi-axis Differential Optical Absorption Spectroscopy (MAX-DOAS) at Eureka, Canada.

The GOME-2 tropospheric BrO columns used in this study are described in further detail by Blechschmidt et al. (2016). In summary, tropospheric BrO vertical columns (VCD$_{trop}$) were obtained based on the approach of Begoin et al. (2010) for deriving BrO total slant column densities by the DOAS (Platt et al., 1994) method using a 336-347 nm fitting window (Afe et al., 2004) and on Theys et al. (2011) for stratospheric correction. The latter involves the use of a climatology of stratospheric vertical column densities (VCDs) of BrO estimated by the BASCOE chemical transport model (Errera et al., 2008; Viscardy et al., 2010). The stratospheric VCDs were converted to slant columns by application of a stratospheric air mass factor and

then subtracted from total slant columns. A tropospheric air mass factor was applied for conversion to $VCD_{trop}$ assuming that all BrO is located and well mixed within the lowermost 400 m of the troposphere over ice or snow with a surface reflectance of 0.9. A sensitivity study for a BEE case showed that the GOME-2 tropospheric BrO column has a moderate sensitivity to the stratospheric BrO column, e.g., a variation in the $VCD_{strat}$ of 15–30% leads to a change in $VCD_{trop}$ of about 0.5 to

5 $1\times10^{13}$ molecules $cm^2$, respectively (Zhao et al., 2016). The influence of clouds on GOME-2 BrO retrievals and the implications for studying bromine explosion events using GOME-2 data are discussed in Blechschmidt et al. (2016). GOME-2 tropospheric BrO column maps (0.5x0.5 degree grid) and time series based on subsetted data of $VCD_{trop}$ (all measurements having their centre within a distance of <40 km from the ground-station) at Resolute, Eureka and Alert sites are used here.

MAX-DOAS measurements were performed at the Polar Environment Atmospheric Research Laboratory (PEARL) Ridge
Laboratory (610 m) in Eureka. Spectra were recorded in the UV using a grating spectrometer (1200 groves/mm grating) with a cooled (200 K) charge-coupled device (CCD) detector at 0.4-0.5 nm resolution. Elevation angles of 30°, 15°, 10°, 8°, and 5° (6° in 2011) were used in the elevation scans, and measurements were only taken with solar elevation above 4°. Differential slant column densities (dSCDs) of BrO and the oxygen dimer ($O_4$) were retrieved using the settings described in Zhao et al. (2016). Reference spectra for the DOAS analysis were interpolated from zenith measurements taken before and after each
elevation scan. dSCDs were converted to profiles using a two-step optimal estimation method (Frieβ et al, 2011). First, aerosol extinction profiles were retrieved from $O_4$ dSCDs, and then the extinction profiles were used as a forward model parameter in the BrO retrieval. The retrievals were performed on a 0-4 km altitude grid with 0.2 km resolution. Due to the altitude of the instrument (610 m) and the lack of low or negative elevation angles, the retrieved profiles are only sensitive to well-mixed BrO in a deep boundary layer, and to lofted BrO events.

**3 Models**

A global chemistry transport model, p-TOMCAT, and a global chemistry climate model, UKCA, are used in this study. The offline p-TOMCAT used 6-hour ERA-Interim dataset to drive its winds, temperature and moisture. The ERA-Interim data were taken from the European Centre for Medium-Range Weather Forecasts (ECMWF) (Dee et al., 2011). In this study, a nudged UKCA version is used to ensure a model meteorological field close to the real situation for data-model comparison.
We follow the work of Telford et al. (2008) with a standard nudging relaxation parameter $G=1/6$ $h^{-1}$, which value lies within the range of relaxation parameters used by other models (Jeuken et al., 1996; Hauglustaine et al., 2004; Schmidt et al., 2006). We used the 6hrly ERA-Interim winds and temperature to constrain UKCA model's dynamical field. However, nudging is not applied to all levels; no nudging being applied above level 50 (~48 km), or below level 12 (~2.9 km (the actual height varies depending on the orography). To avoid instability of the model, moisture is not nudged to reanalysis data, therefore it is free
running.

Both models applied a non-local boundary layer mixing scheme, but p-TOMCAT based on the parameterisation of Holtslag and Boville (1993), while UKCA based on the scheme of Lock et al. (2000). In terms of convective mass flux, p-TOMCAT applied the scheme of Tiedtke (1989) – which has been updated to increase convective transport to the mid and upper troposphere (Barret et al., 2010; Feng et al., 2011), and UKCA applied the bulk convection model of Gregory and Rowntree
(1990). As shown in a multi-model inter-comparison in tropics, these two models showed different behaviour in terms of deep convective transport of tropical boundary layer tracers (Hoyle et al., 2011). The clouds and precipitation schemes are also different between the two models (Russo et al., 2011), resulting in different wash-out rates for aerosols and soluble chemical compounds. The precipitation bias in the op-TOMCAT model (Giannakopoulos et al., 2004) is remedied by applying a correction to force the simulated precipitation values towards Global Precipitation Climatology Project (GPCP) observations
(Adler et al., 2003), following the work in Legrand et al. (2016). This corrected precipitation scheme has been used in recent

sea salt aerosol modelling works (Rhodes et al., 2017; Yang et al., 2019). However, precipitation in UKCA is free running, therefore the two models may have different wet removal rates for soluble gaseous-phase species. Details of other model configurations, mainly in chemistry scheme used are described in sections 3.1 for p-TOMCAT and 3.2 for UKCA.

In addition to the two global chemistry models, we used back-trajectories from the NOAA Hybrid Single-Particle Lagrangian Integrated Trajectory (HYSPLIT) model (Stein et al., 2015; Rolph et al. 2017) for air-mass history study of the selected ODE case in section 4.2.

## 3.1 p-TOMCAT model

The Cambridge parallelised-Tropospheric Offline Model of Chemistry and Transport (p-TOMCAT) has a horizontal resolution of $2.825° \times 2.825°$ (longitude $\times$ latitude) and 31 vertical layers from the surface to about 10 hPa (~31 km) at the top layer. Sea-ice coverage and sea surface temperatures are monthly and taken from the Hadley Centre Sea Ice and Sea Surface Temperature dataset (Rayner et al., 2003). The p-TOMCAT non-local vertical diffusion scheme is taken from the National Centre for Atmospheric Research Community Climate Model, Version 2. This scheme determines the planetary boundary layer (PBL) height explicitly and takes account of large-scale eddy transport that can occur throughout the boundary layer even when part of it is statically stable. Implementation and validation of the PBL scheme was carried out by Wang et al. (1999). The model behaviour in terms of vertical mixing of atmospheric tracer and air mass transport has been reported in Russo et al. (2011) and Hoyle et al., (2011).

The ozone photochemistry scheme applied to the model has been detailed in previous studies (Law et al., 1998, 2000) and Savage et al. (2004), with updates including an isoprene chemistry scheme, same as the one implemented to the UKCA model by Young et al. (2009) according to the method of Poschl et al. (2000), a hydrolysis reaction of $N_2O_5$ on aerosols and cloud droplets (Yang et al., 2005), a tropospheric bromine scheme involving both gaseous-phase reactions (Yang et al., 2005) and heterogeneous reactions (Yang et al., 2010), and a Fast-J photolysis scheme developed by Voulgarakis et al. (2009b), which is not used in this study. They found that $N_2O_5$ hydrolysis can cause net $NO_X$ loss at high latitudes by up to 60% in the northern hemisphere and ~80% in the southern hemisphere (Yang et al., 2005). They found that including halogen-related heterogeneous reactions on aerosols and cloud droplet can significantly increase polar BrO partitioning by a factor of ~3 (Yang et al., 2010). This heterogeneous reaction scheme for halogen reactivation was also implemented to the UKCA model (Yang et al., 2014; Dennison et al., 2019; Ming et al., 2020).

Ozone is dry-deposited in the bottom model layer with dry deposition velocity inferred from the study of Ganzeveld and Lelieveld (1995) by Giannakopoulos (1998). The original dry deposition velocity over ocean and snow (=0.05 cm s$^{-1}$) is reduced to 0.01 cm s$^{-1}$ in this study following recent modelling work by Hardacre et al. (2015) and Luhar et al. (2018) as well as Helmig et al. (2007). Since p-TOMCAT only covers part of the stratosphere with a top layer height of ~31 km, a simplified stratospheric chemical scheme has to be used, including a pre-prescribed top boundary condition for ozone. Therefore, p-TOMCAT model is quite different from the UKCA model in upper troposphere and lower stratosphere. However, it is unlikely that the downwards transport of air mass in the polar region may significant influence near surface bromine. Recent change to p-TOMCAT the tropospheric halogen chemistry scheme includes updates to dry and wet deposition schemes as reported in Legrand et al. (2016). Tropospheric bromine comes from three emission sources: (i) very short-lived substances (VSLS) bromocarbons following the work of Warwick et al. (2006) with reduced flux for $CH_2Br_2$ (Yang et al., 2014), (ii) Open ocean sea spray (Yang et al. 2005, Breider et al., 2009; Parrella et al., 2012), and (iii) sea-ice-sourced SSA in polar regions following the work of Yang et al. (2008; 2010; 2019). Here we define total inorganic bromine $Br_Y = HOBr + HBr + BrO + Br + BrONO_2 + BrNO_2 + 2*Br_2 + BrCl$.

A process-based SSA transport, dry and wet deposition scheme has been implemented in the model by Levine et al. (2014) based on the work of Reader and McFarlane (2003). The open ocean sea spray emission scheme follows Jaeglé et al., (2011),

and the sea-ice-sourced SSA scheme follows the latest work of Yang et al. (2019). Both open-ocean-sourced and sea-ice-sourced SSA (denoted as OO and SI, respectively) are tagged in 21 size bins covering dry NaCl diameter of 0.02-20 μm in order to track their history for online calculation of their surface density for heterogeneous reaction rates.

All parameters applied in this study for the Arctic SSA simulation are directly taken from our recent SSA modelling work by

5 Yang et al. (2019), including a 3.5 times Antarctic snow salinity for the Arctic. The Antarctic Weddell Sea cruise data (Frey et al., 2019) is a probability of surface snow salinity, which is different to the constant salinity value (=0.3 psu, practical salinity unit) used in Legrand et al. (2016), Zhao et al. (2017) and Rhode et al. (2017). The trebled snow salinity assumption is taken from Yang et al. (2008) to reflect the likelihood that Arctic snow is more saline than in the Antarctic due to reduced precipitation. This assumption is partly justified by surface snow [Cl-] concentrations observed in the two poles. For instance,

an averaged surface snow (top 1-2 cm) [Cl-] concentration of 368 μM is derived from the Weddell Sea, Antarctic (https://ramadda.data.bas.ac.uk/repository/entry/show?entryid=853dd176-bc7a-48d4-a6be-33bcc0f17eeb, Frey et al. (2019)). In the Arctic, Pratt et al. (2013) reported a mean surface [Cl-] concentration of 1,121 μM (top 1 cm) over coastal sea ice near Barrow, Alaska, and Krnavek et al. (2012) reported a much higher surface [Cl-] concentration of 21,058 μM over first-year sea ice and 63,217 μM over multi-year sea ice over a slightly deeper depth of 2~3 cm below the surface. They are about 3, 57

and 172 times of the Weddell Sea surface salinity. The relative higher salinity in the Arctic is partly related to less precipitation as already mentioned. For instance, the depth of snowpack on sea ice near Barrow, Alaska is in a range of 10-40 cm (Krnavek et al. (2012), while in the Weddell Sea, the mean snow depth over FYI is 20.9 cm and 50.0 cm over MYI (Frey et al., 2019). Other parameters used in this study include a mean snow age of 3 days for the Arctic following the recent work of Huang and Jaeglé (2017). We assume that the evaporation rate of blowing snow particles is controlled by the moisture gradient between

the surface of the particle and the ambient air, an evaporation function of $\frac{dm_i}{dt}=d_i$, with $m_i$ being water mass and $d_i$ being diameter of snow particle) i.e., the classic mechanism in Yang et al. (2019). For blowing snow size distribution, we used a shape parameter $\alpha$=3 and a scale parameter $\beta$=37.5 μm, with a SSA production ratio N =20 (i.e., 20 SSA particles formed from one saline wind-blown snow particle during sublimation). This set of parameter corresponds to the SI_Classic_B×20 run in Yang et al. (2019) and is one of the best parameter sets that matched the Weddell Sea SSA in size range of 0.4 – 10 μm. Also,

this set gave the highest SSA mass loading in polar regions (Yang et al., 2019).

To parameterize bromide release from SSA in the Arctic, two different patterns of bromine depletion factor (DF) are used. Table 1 contains a seasonal DF scheme with a maximum value of 0.53 in May and a minimum of 0.07 in December. This seasonal scheme is derived from the bulk SSA bromide depletion strength from Dumont d'Urville (Legrand et al., 2016), Cape Grim and Macquarie Island (Ayers et al., 1999) in the Southern Hemisphere with a six-month shift of the phase in order to

30 apply to the NH. Since similar year-round in situ dataset from the Arctic is not available, we could not justify this seasonal DF pattern, which demands further systematic measurements in the Arctic. As used in previous modelling studies, a size-dependent (non-seasonal) DF scheme for the NH is used for comparison (Supplementary Table 1), which is derived from previous work of Yang et al. (2008, 2010) and Breider et al. (2009). Note that we simply apply these DF schemes to all SSA emitted and do not distinguish between the open-ocean-sourced and the sea-ice-sourced SSA in terms of bromide release. However, this

approach may introduce bias as freshly emitted sea spray is alkaline with pH>8 and needs acidification first by absorbing sulphate or nitrate before bromide can be liberated to the atmosphere through heterogeneous reactions (e.g., Breider et al., 2009). In contrast, snowpack in the Arctic is largely acidified with pH of 4-6 due to local acidity contamination (e.g., de Caritat et al., 2005). The difference in initial conditions between sea spray and sea-ice-sourced SSA may affect bromide release in both timing and strength, which has not been considered by our models. Thus, we may overestimate the open-ocean-sourced

SSA effect in polar regions, as the alkaline buffering effect is not considered.

## 3.2 UKCA

UKCA, a version of the UK Earth System Model with Chemistry and Aerosols, has a dynamical core from the Met Office Unified Model (UM) (Morgenstern et al., 2009). A nudged model version-7.3 is used in this study with a horizontal resolution of $3.75° \times 2.5°$ and 60 vertical layers from the surface to ~84 km. The tropospheric chemistry scheme was built on the scheme in p-TOMCAT model, but contains a comprehensive stratospheric chemistry scheme for climate studies (Braesicke et al., 2013; Banerjee et al., 2014; Ming et al., 2020). In terms of SSA production, the same schemes for open ocean sea spray and for sea-ice-sourced SSA as in the p-TOMCAT are used, apart from the fact that the SSA in UKCA runs is no longer being tagged and tracked for online calculation of heterogeneous reaction rates. Therefore, the emitted SSAs are just used for bromide emission. For heterogeneous reactions, the aerosol surface area density is calculated using the archived monthly climatology aerosol dataset taken from the CLASSIC scheme (Johnson et al., 2010). In p-TOMCAT model, heterogeneous reactions occur also on cloud droplets, but UKCA does not include such reactions on cloud droplets. Therefore, in free troposphere, the BrO partitioning in UKCA may be lower than that in p-TOMCAT, which may result in more soluble inorganic bromine species being washed-out by precipitation in UKCA, as discussed in section 4.

Note that UM-UKCA is a complex chemistry-climate coupling model, covering the whole atmosphere including both troposphere and stratosphere. In many aspects of dynamics and chemistry, it behaves quite differently from the p-TOMCAT CTM. A detailed comparison of model characteristics in vertical mixing and transport of tropical boundary layer tracers was performed by Russo et al. (2011) and Hoyle et al. (2011). The bottom model layer of UKCA, in which chemical compounds such as ozone undergo dry deposition, is ~20 m thick, while in p-TOMCAT it is ~60 m thick. All released SSA and bromine (in the form of $Br_2$) are put in the bottom model layer before they are further vertically mixed and horizontally transported. These differences in model vertical resolution may affect model output even if other factors are the same. Although the two models are quite different, e.g., in absolute values of chemical compounds, the relative changes in response to changes of bromine loading, for example, are still informative and will be our major interest and focus of the discussion.

## 3.3 Model experiments

Table 2 lists major model experiments performed in this study. The two model base runs, pTOMCAT_control and UKCA_control, contain reactive bromine emissions from both sea-ice-sourced and open-ocean-sourced SSA and VSLS bromocarbons. The pTOMCAT_No_Br run does not include any bromine emission, therefore, it is a model run without bromine chemistry. The pTOMCAT_VSLS and UKCA_VSLS runs only contain bromocarbons as a source of reactive bromine (without bromine from open ocean and sea ice). The pTOMCAT_SI_VSLS and UKCA_SI_VSLS runs only contain sea-ice-sourced SSA and bromocarbons as sources of reactive bromine. Similarly, the pTOMCAT_OO_VSLS and UKCA_OO_VSLS runs only contain sea-spray-sourced SSA and bromocarbons as sources of reactive bromine. By checking the differences between experiments from the same model, we expect to separate individual bromine source contributions to Arctic boundary layer bromine mixing ratio and ozone mixing ratio. Similarly, by checking the differences between the two model responses, we will see model-induced uncertainty, e.g., in both the bromine mixing ratio and ozone mixing ratio. This is because both models employ very similar bromine emissions. Therefore, the differences are mainly due to different model configuration either in their physical aspect, including precipitation, boundary layer dynamics, land use, etc, or in their chemistry aspect involving key atmospheric species such as ozone or OH, and heterogeneous reactions.

Apart from the pTOMCAT_Fixed_DF run, in which a fixed (non-seasonal) DF scheme is used (Table S1), all model experiments apply the same bromine DF scheme shown in Table 1. A multiple-year integration (2006-2008) is performed, with averaged outputs used as a climatology for comparison. Several spring runs in 2010, 2011 and 2013 are made with more frequent outputs for ODE and BEE comparisons: 1-hourly output frequency in pTOMCAT_control and 3-hourly output frequency in UKCA_control are used for further analysis and model-data comparisons. These years are selected because either significant ODEs or BEEs are observed at one or more sites.

To investigate model sensitivity to key parameters, such as snow salinity, DF, cut-off size and SSA spectrum, we performed additional model experiments (Table 3) with a range of uncertainty for each parameter. For example, pTOMCAT_high_salinity applies a 10 times Weddell Sea snow salinity, and pTOMCAT_low_slinity applies a 1 times salinity; pTOMCAT_2×DF applies a doubled DF and pTOMCAT_0.5×DF applies a halved DF; pTOMCAT_SSA20μm applies a large cut-off threshold with dry NaCl radius of 20 μm and pTOMCAT_SSA5μm applies a small cut-off threshold of 5 μm; pTOMCAT_spectrum_1 applies same parameters as in the control run but a small N=10 and pTOMCAT_spectrum_2 applies a different parameter set with N=1 and different $a$, $\beta$ and $\frac{dm_i}{dt}$ function (see Table 3), which corresponds to the SI_Base run in Yang et al., (2019). This sensitive experiment is only integrated for one year (2007) with results are compared to the pTOMCAT_control run result, as discussed in section 5.

## 4 Results

### 4.1 Surface ozone seasonality

Figure 2 shows observed monthly mean surface ozone VMRs at the six Arctic locations, two inland (Summit and Pallas) and four coastal (Alert, Barrow, Tiksi and VRS) sites. Spring ozone maxima of >50 ppbv at Summit and ~45 ppbv at Pallas were observed. However, at coastal sites, spring-time ozone is depleted, with low VMRs of 15~20 ppbv, which are comparable to or even lower than their summer ozone minimum in July-August. The summer minimum is thought attributable to enhanced ozone photo-dissociation, where $NO_X$ levels are low, and increased dry deposition to plants (e.g., Hatakka et al., 2003; Engvall Stjernberg et al., 2011). At higher latitude sites such as Alert and VRS, that are within the polar dome and surrounded by Arctic tundra with sparse vegetation, there is normally still snow coverage even in mid-summer, so the local effect of dry deposition to plants may not be as significant as at Pallas or other sites located further south. However, the long-range transport from lower latitudes of ozone affected by summer plants may result in vegetation having an effect on these sites. For example, the suppressed high latitude summer ozone in Siberia is related to deposition loss to vegetation during long-range transport into the Arctic (Engvall Stjernberg et al., 2011). Figure 2 also shows that model runs without bromine chemistry (pTOMCAT_No_Br) and with bromocarbons only (pTOMCAT_VSLS) can generate spring ozone maxima at all six sites. When open-ocean-sourced reactive bromine is included (orange line in Fig. 2), the spring ozone peak is reduced significantly. The OO-sourced bromine can cause ozone reductions in all seasons (Fig. 3), with a maximum reduction of > 10 ppbv in April and a minimum reduction (1~2 ppbv) in summer. However, the OO-sourced reactive bromine does not alter the ozone seasonality pattern, as the spring ozone peak remains (Fig. 2). On the other hand, sea-ice-sourced reactive bromine (red line in Fig. 2) can significantly perturb the ozone seasonal cycle, by removing the spring ozone peak completely. On average, SI-sourced bromine can cause a maximum ozone loss of > 10 ppbv in April at coastal sites (Fig. 3), similar to the OO-sourced reactive bromine effect. In autumn, SI-sourced bromine only weakly influences the bromine budget and ozone loss. Model runs which contain bromine sources from both OO and SI (black line in Fig. 3) can cause a peak of annual ozone loss of > 25 ppbv (in monthly mean) in April at coastal sites, giving the best match to the observations (Fig. 2). However, inclusion of halogen chemistry leads to severe underestimation of spring ozone at Summit and Pallas.

A similar effect of the SI- and OO-sourced reactive bromine on surface ozone can be seen in UKCA runs but with net ozone loss only half of that seen in p-TOMCAT runs (Fig. S1). As discussed below, this difference is consistent with the difference in total inorganic bromine $Br_Y$ between the two models: a spring surface layer $Br_Y$ maximum of 10-30 pptv is simulated in pTOMCAT_control (Fig. 3), which is about twice that (5-10 pptv) in UKCA_control (Fig. S1). This is consistent with zonal mean (April) $Br_Y$ differences between the two models as shown in Fig. S2. Since both models employ a very similar bromine emission flux (e.g., SSA production driven by ECMWF data and with the same bromine depletion factor), the difference in $Br_Y$ between the two models is likely due to the difference in removal process of inorganic bromine species, such as HBr, HOBr, $Br_2$ and $BrONO_2$, which are either dry and/or wet deposited. Previous model simulations have shown that, on the global

scale, precipitation wash-out is responsible for ~90% of the removal of tropospheric $Br_Y$ (Yang et al., 2005). In polar regions, where the precipitation rate is relatively low, dry deposition to the surface is another efficient pathway for inorganic bromine removal in surface layer., The different approaches in chemical scheme applied by the models may also affects inorganic bromine deposition rate through influencing partitioning of inorganic bromine species. This is because some species (e.g.,
HBr, HOBr) are very soluble while others are not (e.g., BrO). A higher BrO partitioning is expected at a higher ozone concentration, and vice versa. Therefore, an overestimated ozone is expected to have a negative feedback to bromine removal and net ozone loss via bromine chemistry. In addition, p-TOMCAT considers heterogeneous reactions on cloud droplets while UKCA does not, this difference may explain why BrO partitioning in p-TOMCAT is higher than that in UKCA, especially in free troposphere, where BrO partitioning can be as large as 50% (Fig. S2). In addition, the higher BrO partitioning in p-
TOMCAT also attribute less $Br_Y$ removal by dry and wet depositions.

Comparing the surface layer BrO/$Br_Y$ ratio between the two models (Fig. S3), we can see that both UKCA_control and pTOMCAT_control give a very similar spring peak with a ratio around 20-30% at coastal sites, with an exception at VRS, where a ratio of up to >60% is simulated in the UKCA_control run. The largest discrepancy appears in summer at some costal sites such as Alert, Tiksi and VRS, where a second summer peak ratio is simulated in the UKCA_control, which is likely
attributed to the obviously overestimated summer ozone concentrations by this model (Fig. S4).

From Fig. 2 and Fig. S1 we can see that, on average, surface BrO VMRs at inland sites are smaller than that at coastal sites in both model outputs. For example, in April, mean BrO is ~1 pptv at Summit and ~0.5 pptv at Pallas, at coastal sites, VMRs are between 2 and 7 pptv in pTOMCAT_control run. In terms of $Br_Y$, as shown in Fig. 3, in April, both OO- and SI-sourced bromine contributes roughly the same amount (6~8 pptv) at the two inland sites of Summit and Pallas. At coastal sites (Alert,
Barrow, Tiksi and VRS), the OO-sourced bromine contributes one-sixth to half of the SI-sourced $Br_Y$, i.e., 4~5 pptv vs 8~30 pptv. The large (and small) gradient in $Br_Y$ of the SI-sourced (and the OO-sourced) bromine between inland and coastal sites indicates that SI-sourced bromine is locally sourced, while OO-sourced bromine is remotely transported and thus has a smaller horizontal gradient in VMR. In addition, VSLS bromocarbons only have a relatively small contribution of ~0.5 pptv $Br_Y$ in spring-summer (Fig. 3), corresponding to an ozone loss of ~1 ppbv. As shown in Fig.S7, VSLS contribution to tropospheric
$Br_Y$ over the Arctic increases from near surface layer ~0.5 pptv (in April and July) to ~2 pptv at ~200 hpa. In spring (April), it only accounts for 2~4% of the total $Br_Y$ in the surface layer and ~40% at 200 hpa; in July, it accounts for 15~20% in the surface layer and >60% at 300~400 hpa.

## 4.2 Surface ozone frequency distribution

Figure 4 shows NH summer (July) surface ozone frequency distribution at four coastal sites from both observation and p-
TOMCAT runs. Climatology clearly indicates a single ozone peak distribution with peak VMRs around ~20 ppbv. The p-TOMCAT model successfully reproduces this summer single peak distribution frequency, though it overestimates it by a few ppbv, e.g., at Alert and VRS. Similar to what is reflected in Fig. 3, the bromine effect on ozone loss is small, only 1-2 ppbv. In April, the observed ozone distribution frequency is quite different from the summer pattern, as a flat distribution across ozone bins is observed with a large ozone depletion fraction at ozone VMRs <10 ppbv (Fig. 5 and 6). Although both p-
TOMCAT (Fig. 5) and UKCA (Fig. 6) fail to reproduce this flat distribution pattern, the two model runs with SI-sourced bromine implemented can largely reproduce ozone depletion fraction (at ozone < 10 ppbv). It is interesting to note that though OO-sourced bromine alone can cause April monthly mean ozone to drop by 5~10 ppbv at coastal sites (dashed blue line vs solid blue line in Fig.5 and 6), but itself alone cannot generate any ozone depletion at VMRs < 20 ppbv, indicating that this remotely sourced bromine to the Arctic is not responsible for coastal ODEs; rather, it only affects background ozone. On the
other hand, SI-sourced bromine can cause ozone depletion with a significant fraction of ozone VMRs < 10 ppbv (dashed orange line in Fig. 5 and Fig. 6), supporting the suggestion that locally sourced bromine (from sea ice) is responsible for spring ODEs.

As discussed in Section 4.3, the failure of models in reproducing the flat ozone distribution in spring is likely attributable to the coarse resolution of the models used. For instance, the ~2.8°×2.8° in p-TOMCAT and 3.75°×2.5° in UKCA horizontal resolution means that any sub-grid scale events will not be captured and represented by the model, and a finer resolution model may be needed to have better representation of the observations.

### 4.3 Spring ozone depletion events

### 4.3.1 Time series

Figure 7 and 8 show month-long time series of surface ozone at Tiksi (May 2011), Barrow (March 2010), Summit (April 2010), VRS (April 2010) and Alert (April 2010) in observations (black lines), along with pTOMCAT_control output (fig. 7) and

10 UKCA_control output (Fig. 8) of surface ozone shown (in red) and BrO (in blue) from the nearest gridbox. Also shown in Fig. 7 and 8 are maximum and minimum ozone taken from five adjacent gridboxes in the bottom model layer of each model with the aim of investigating the effect of model resolution on output. From Fig. 7, we see that both the p-TOMCAT and UKCA models can largely reproduce large ODEs, e.g., the 1-week-long ODE during 7-11 May 2011 at Tiksi and 10-15 March 2010 at Barrow, and the 3-day long ODE during 22-24 April 2010 at VRS. However, the model has a very limited ability to represent

small-scale events that last from a few hours to ~1 day. Large discrepancies are found in both timing and magnitude of the ozone depletion. For instance, the one-day long ODE on ~22 April 2010 at Alert is not well captured by the central gridbox closest to the site in both models. For example, the p-TOMCAT simulated minimum ozone occurs later by about one day. However, this observed ozone minimum is reproduced if adjacent gridbox results are taken into account. In general, the pTOMCAT_control run central gridbox surface ozone is significantly correlated with observed ozone with medium to high

correlation coefficients $R$ of 0.68 at Tiksi, 0.49 at Barrow, 0.60 at Summit and 0.75 at VRS and a small $R=0.22$ at Alert ($p<0.001$ at all sites). (Fig. 7 and Table 4 column 2). UKCA_control run shows a similar result with different correlation coefficients $R$ of 0.68 ($p<0.001$) at Tiksi, 0.18 ($p<0.01$) at Barrow, 0.47 ($p<0.001$) at Summit, 0.62 ($p<0.001$) at VRS and 0.41 ($p<0.001$) at Alert (Fig. 8).

Figure 7 and 8 show that large ODEs are mostly accompanied by enhanced BrO. Statistical analysis shows (for the p-TOMCAT

result only) that surface BrO simulated is negatively correlated with observed ozone at Tiksi, Summit and VRS with R of -0.49 ($p<0.001$), -0.19 ($p<0.001$) and -0.51 ($p<0.001$) respectively (Table 4 column 3), while at Barrow and Alert, the correlation is not significant (with R of ~-0.05). However, the correlation between observed surface ozone and simulated tropospheric BrO column becomes significant with $R$ of ~-0.2 ($p<0.001$) (Table 4 column 4) at these two sites; a similar phenomenon is seen between observed ozone and GOME-2 $BrO_{trop}$ at Tiksi, Barrow and VRS (Table 4 column 5), though this

correlation does not exist at Summit and Alert. In general, boundary layer ozone is influenced by column BrO in the low troposphere rather than by surface BrO, though these two factors are largely correlated in modelling output (Table 4 column 8). For example, observed ozone at Barrow and Alert significantly correlated with simulated tropospheric column BrO but surface BrO. This is because ozone has much longer lifetime than BrO, thus through vertical mixing and/or air ventilation at top of the boundary layer, ozone and BrO in the free troposphere may influence surface ozone within the boundary layer. For

a specific location, surface ozone may not always represent ozone levels in higher layers, so is surface BrO. Therefore, ozonesonde vertical profile data may supply more information than surface data. Note that at extremely low ozone conditions (e.g., after a complete ozone consumption by halogen chemistry in a stable boundary layer), the negative correlation between BrO and ozone concentrations may not exist (Zhao et al., 2016). This is because under that condition, the photochemical equilibrium is shifted from BrO towards atomic Br.

Figure 9 and 10 show time series of tropospheric column BrO from GOME-2, along with outputs from pTOMCAT_control (Fig. 9) and UKCA_control (Fig. 10). The GOME-2 $BrO_{trop}$ data are tropospheric vertical column BrO for each site at the overpass time. In general, p-TOMCAT $BrO_{trop}$ matches GOME-2 $BrO_{trop}$ well with a correlation coefficient $R=0.59$ ($p<0.001$)

at Tiksi, 0.37 ($p<0.001$) at Barrow, 0.33 ($p<0.001$) at VRS and 0.30 ($p<0.001$) at Alert. The lowest correlation is seen as Summit with $R=0.16$ ($p<0.1$), where the satellite column does not show significant day-to-day perturbation (Table 4 column 7 or Fig. 9). In UKCA, a similar correlation is found with $R=0.32$ ($p<0.01$) at Tiksi, 0.31 ($p<0.01$) at Barrow, 0.39 ($p<0.001$) at VRS. However, at Summit and Alert the correlation coefficients are very small (0.05 at Summit and 0.1 at Alert). p-
TOMCAT model tends to overestimate satellite column BrO data by factors of ~2 during BEEs, e.g., during 7-11 May 2011 and during 9-15 March 2010, when there are large ODEs observed. But during non-ODEs (or non-BEEs) periods, modelled column BrO in p-TOMCAT is in good agreement with the GOME-2 data. In contrast, UKCA BrO$_{trop}$ significantly underestimates during non-BEEs periods, e.g., by a factor of ~10 over Summit, though works well during BEEs. On average, UKCA BrO$_{trop}$ is lower than the observation by ~50%.

**4.3.2 Vertical profiles**

A large ODE observed at Eureka on 3-7 April 2011 has been reported by Zhao et al. (2016), who pointed out that this ODE was a transported event associated with a strong cyclone originating in the Chukchi Sea on 31 March 2011. GOME-2 BrO$_{trop}$ images (Fig. S5) clearly indicated a large spiral BrO plume over the Chukchi Sea on 1 April 2011 (Blechschmidt et al., 2016), which was transported across the Canadian high Arctic in the following days. This event might have influenced both Resolute
and Alert, which are located within ~500 km of Eureka. HYSPLIT 6-day back trajectories ending at 12:00 UTC on 5 April 2011 (Fig. 11) show that the air-mass history of the three sites has a very similar transport pattern, further indicating that these three sites were influenced by the same synoptic system. For this reason, we extend this case study by looking at ozonesonde data from all three sites as shown in Fig. 12a-c and 13a-c. Ozonesonde data clearly indicate a severe ozone depletion layer at altitude < 2 km during 3-7 April 2011 at both Resolute and Eureka, with minimum ozone less than 1 ppbv in the near-surface
layer. At Alert, the ozone depletion strength was a bit weaker than at the other two sites, but the depleted ozone layer still reaches an altitude of 1~1.5 km. Moreover, the most severe ozone depletion at Alert on 4 April is not near the surface, but rather at an elevated height of 500-800 m. The simulated ozone profiles in UKCA_control run (Fig. 12d-f)) generally match the ozonesonde profiles in the 0-4 km range. For this ODE, both the timing of occurrence and height range of the ODE are roughly captured by the UKCA model. The modelled BrO profile is shown in Fig.12g-i, with enhanced BrO being simulated
in association with ozone depletion. At Eureka, the simulated maximum surface BrO is similar to the MAX-DOAS measurement with a maximum VMR of ~20 pptv for this event (Zhao et al., 2016; 2017). In pTOMCAT_control run, ozone-poor air is not limited to near surface layer (or < 1.5 km), rather depleted ozone spread is well spread over the lower troposphere to a height of ~4 km (Fig. 13d-f). Similarly, simulated BrO in p-TOMCAT is also uniformly spread in lower troposphere (Fig. 13g-i). A zoom-in comparison between the GOME-2 tropospheric column BrO and model-integrated BrO for the period of 1-
10 April 2011 is shown in Fig. 14 (for UKCA) and Fig. 15 (for p-TOMCAT). Satellite BrO columns reached a peak firstly on 3 April 2011 at Resolute. One day later, the peak appeared at Eureka and Alert. The UKCA_control run shows a similar transport pattern of enhanced BrO, which reached Resolute first, and then Eureka and Alert (Fig. 14). However, the pTOMCAT_control run shows enhanced BrO reaches Eureka and Resolute first and then Alert later (Fig. 15). The differences are likely related to the different model resolutions and gridbox coordinate. The above finding is consistent with our previous
conclusion made in Zhao et al. (2016) that this ODE is transported associated with a large cyclone.

A strong correlation between the modelled BrO$_{trop}$ and the GOME-2 BrO$_{trop}$ data can be seen at Resolute with a high $R$ of 0.71 ($p<0.001$) in UKCA (Fig. 14d) and 0.59 ($p<0.001$) in p-TOMCAT (Fig. 15d). At the other two sites, the correlation is positive in UKCA but with small $R$ of ~0.2, in p-TOMCAT, the $R$ is small to medium: 0.45 at Eureka ($p<0.01$) and 0.33 ($p<0.05$) at Alert. The UKCA BrO$_{trop}$ is on average lower than the satellite data by ~50% (refers to the regression equations shown in Fig.
14), which is opposite to the overestimated BrO$_{trop}$ by 50~90% in p-TOMCAT model (in Fig. 15). This difference is in line with the above discussed differences in total Br$_Y$ between the two models (in Section 4.1), although the same total bromine emissions are applied.

## 5 Discussions

Regarding tropospheric total $Br_Y$ in the Arctic, as mentioned previously, the model-to-model difference can be as large as 100% at near surface layer (under the same bromine loading). As a consequence, the ozone loss due to bromine chemistry can be different by a factor of two. The relatively high $Br_Y$ in pTOMCAT_control run is partly due to the higher BrO partitioning in p-TOMCAT (attributed to the inclusion of heterogeneous reactions on cloud droplets) and thus less wet removal of soluble bromine species, and partly due to stronger vertical mixing of air masses in lower troposphere and thus less dry deposition removal of reactive bromine species from the surface layer.

On a global scale, the uncertainty of the sea spray (from open ocean) source can be a factor of four (Lewis and Schwartz, 2004). On sea ice, the blowing snow related SSA production is sensitive to both snow salinity and bulk sublimation flux calculated (as a complex function of near surface wind speed, temperature and relative humidity, etc) (Yang et al., 2019). Although we lack snow data on Arctic sea ice to strictly constrain the 3.5 times Antarctic Weddell Sea snow salinity used for the Arctic (Section 3.1), the likelihood of higher snow salinity in the Arctic implies that there is more SSA generated from same amount of blowing snow sublimation flux (also with slightly larger SSA in size). As a consequence, there is more reactive bromine released from blowing snow in the Arctic than in the Antarctic. Given that the snow salinity effect on SSA mass production is almost linear, the uncertainty caused by this factor can be linearly estimated when more snow data in the Arctic are available. However, in terms of relative bromine release from SSA, the actual emission flux varies and depends on the salinity, this is due to the cut-off threshold size applied (i.e., a dry NaCl radius of 10 µm in the control run). Therefore, the reactive bromine release from SSA is a function of snow salinity and SSA spectrum.

Another factor that may directly affect reactive bromine emission is the depletion factor. Fig. S6 shows simulated ozone from the pTOMCAT_Fixed_DF run, in which a fixed bromine DF scheme (Table S1) is used. For comparison, the pTOMCAT_control run result is shown, in which the seasonal DF scheme (Table 1) is applied. As can be seen, the timing of the spring ozone minimum shifts slightly from April in pTOMCAT_control towards March in pTOMCAT_Fixed_DF, which makes the model agreement poorer, as the observed ozone minimum is in May at the four coastal sites. To achieve better agreement with the observations, the model needs either an even larger seasonal amplitude of bromine DF than that in Table 1 or a further shift of the DF phase by at least one month, e.g., to allow the annual maximum DF (=0.53) to shift from May to June. However, due to lack of year-round SSA bromide data in the Arctic, we could not validate the DF patterns used in this study as this requires systematic measurements of the SSA bromide depletion strength in the Arctic. This is critical as local SSA is a large source of bromine and the seasonal DF not only affects the timing but also affects the total bromine flux to the atmosphere. Model bias also comes from applying the same depletion factor scheme (i.e., Table 1) to both open-ocean-sourced sea spray and sea-ice-sourced SSA. As we know that freshly released sea spray is alkaline with pH>8, and therefore, the anions in sea spray may buffer the absorbed nitrate and sulphate before getting acidified to allow bromide to be released through heterogeneous reaction, e.g., $HOBr+Br \rightarrow Br_2$ (e.g., Sander et al., 2003; Breider et al., 2009). On sea ice, the situation could be different as surface snow may have been pre-acidified before grains are lifted in to the air to form SSA. Unfortunately, this difference in the process of bromide liberation from SSA particles is beyond the scope of this study, but we note that it could result in bias, e.g., in bromide releasing from air-borne SSA in both strength, timing and locations.

To investigate model sensitivity to the above key parameters used in describing sea-ice sourced SSA and reactive bromine release from SSA, we performed additional model experiments (in Table 3) by altering one or a few parameters in each experiment and comparing the output with the pTOMCAT_control output (for year 2007). For most key parameters, we designed a pair run with one applying a higher value and the other a lower value than in the control run. Model results are shown in figure 16 with derived sea-ice sourced $Br_Y$ (April) and ozone (as well as change with respect to the control run) shown in Table 3.

Since the control run applied a 3.5 times Weddell Sea salinity, the 10 times salinity run in pTOMCAT_high_salinity and 1 times salinity in pTOMCAT_low_salinity is roughly ~3 times and ~1/3 of the control run salinity, respectively. Comparing to

the pTOMCAT_control run, the sea-ice sourced $Br_Y$ (April) in pTOMCAT_high_salinity increases by +94.8%, corresponding to additional ozone loss by -37.5%. Sea-ice sourced $Br_Y$ (April) in pTOMCAT_low_salinity decreases by -60%, corresponding to ozone increase by +61%. It is interesting to note that ozone and $Br_Y$ percentage change in pTOMCAT_low_salinity is at a ratio of 1:1, but in pTOMCAT_high_salinity run the ozone percentage change is only < 1/2 of the $Br_Y$ percentage change. Sea-ice sourced $Br_Y$ in pTOMCAT_SSA20μm (with a large cut-off radius size of 20 μm) increases by +42.3%, corresponding to additional ozone loss by -21.1%, which is almost half of the $Br_Y$ percentage change. On the contrary, sea-ice sourced $Br_Y$ in pTOMCAT_SSA5μm (with a small cut-off radius size of 5 μm) decreases by -55.1%, corresponding to ozone increase by +45.4%. Sea-ice sourced $Br_Y$ in pTOMCAT_2×DF (a doubled DF) increases by 84.5%, corresponding to additional ozone loss by -35.7% (less than 1/2 of the $Br_Y$ change). Sea-ice sourced $Br_Y$ in pTOMCAT_0.5×DF (a halved DF) decreases by -47.3%, corresponding to additional increase by +45.8% (almost same amount of the $Br_Y$ change). Sea-ice sourced $Br_Y$ in pTOMCAT_spectrum_1 (with a small N=10) reduces by -20.1%, corresponding to ozone increase by +16.5%. Sea-ice sourced $Br_Y$ in pTOMCAT_spectrum_2 reduces by -42.2%, corresponding to ozone gain by +38.6%. In all model experiments with reduced $Br_Y$ from sea ice, the percentage change in ozone is almost in the same amount of $Br_Y$ change. However, in the $Br_Y$ increasing cases, the ozone percentage (loss) change is only half or less than that of the $Br_Y$ percentage change indicating that ozone consumption efficiency is getting lower at higher reactive bromine loading, therefore, introducing extra reactive bromine to the environment will not necessary result in equivalent amount of ozone loss as at low reactive bromine loading.

The above model experiments clearly show the possible range of modelled ozone and $Br_Y$ in the Arctic caused by uncertainty of each key parameter involved in the parameterisations. From these runs we can derive the likely maximum effect from the sea-ice sourced SSA from blowing snow. For example, the mean DF values in spring (March, April and May, see Table 1) are ~0.5, a doubling DF indicates all bromide in SSA is released to the air, thus the pTOMCAT_2×DF run represents an extreme scenario with the maximum effect from blowing snow (with other conditions unchanged), so is the pTOMCAT_SSA20μm run, as under this cut-off threshold, almost all SSA formed from blowing snow releases bromide as a source of reactive bromine. pTOMCAT_high_salinity represents another extreme case that shows the large effect from blowing snow. Their combination effect can be multiplied and result in even larger effect. Equivalently, under extremely low snow salinity (such as in pTOMCAT_low_salinty) or small DF (such as in pTOMCAT_0.5×DF), the blowing snow sourced SSA effect on Arctic surface ozone and reactive bromine will be less important than the control run. Therefore, we demand further field measurement to collect data to constrain these key model parameters.

6 Summary For the first time, using two global chemistry models, we have examined the three tropospheric bromine sources (bromocarbons, open ocean sea spray and sea-ice-sourced SSA) and their impacts on Arctic boundary layer bromine and ozone loss. Our modelling experiments show that inclusion of bromine chemistry can greatly improve Arctic surface ozone seasonality reproduction, in particular the spring ozone depletion observed at most Arctic coastal sites, such as Tiksi, Barrow, VRS and Alert. However, inclusion halogen-chemistry leads to severe underestimation of spring ozone at inland sites such as Summit and Pallas. Our model results shows that very short-lived bromocarbons contribution to Arctic tropospheric $Br_Y$ is less than half pptv in the near surface layer, corresponding to small ozone loss of < 1 ppbv. Multi-year simulations show that inclusion of bromine chemistry can cause Arctic surface ozone loss by 10~20 ppbv in spring, with almost half of the ozone loss attributed to open-ocean sourced SSA and the other half from sea-ice sourced SSA. However, without SI-sourced bromine, models cannot reproduce Arctic ozone depletion events, and OO-sourced bromine only affects background atmospheric ozone and cannot by itself produce any polar surface ODEs.

Although a very similar tropospheric halogen scheme applied in the two models, the model-to-model differences are relatively large. For example, boundary layer $Br_Y$ in p-TOMCAT control run is higher than in UKCA control run, which is likely related to the different wet and dry depositions of reactive bromine species. Comparing the GOME-2 satellite data, p-TOMCAT $BrO_{trop}$ overestimates the observations by a factor of ~2 during BEEs, but agrees well with the observations during non-BEEs.

On the contrary, UKCA BrO$_{trop}$ generally underestimates the observations by ~50% during BEEs, but severely underestimates the observation during non-BEEs (e.g., more than an order of magnitude at Summit). Despite the model differences, both model's outputs of time series of surface ozone and tropospheric column BrO (in spring) show significant correlation to the observations at most selected periods, which strongly supports the physical and chemical mechanisms implemented.

Due to the relatively coarse model resolution (e.g., 2~3 degree in horizontal direction), our models cannot resolve small scale ODEs, e.g., with a spatial scale < ~500 km (or with a temporal scale of < ~1 day). Thus, to allow a better reproduction of small-scale ozone events, a fine resolution model is needed. Ozone sonde data from three adjacent high Arctic Canadian sites (Resolute, Eureka and Alert), satellite BrO$_{trop}$ and back-trajectory model output clearly indicate a large ODE (and BEE) in association with a stormy system, which event is successfully captured by the two models, further confirming that ODEs and

BEEs can be long-distance transported. Although our global models cannot be able to reproduce small-scale ODEs, the success of the models in capturing large scale ODEs (and BEEs) gives additional evidence from a chemistry side to the proposed mechanism of SSA production and reactive bromine release from blowing snow on sea ice (Yang et al., 2008; 2019; Frey et al., 2019). Note that the success of the blowing snow mechanism does not necessarily rule out other possibilities, including the proposed candidates of reactive bromine from snowpack, open leads, frost flowers, sea ice surface, etc. Change in sea ice

extent and type in a warming climate will influence Arctic boundary layer chemistry and Arctic climate, including the deposition of atmospheric mercury to the surface (Wang et al., 2019).

**Author Contributions**

XY designed the study, performed model experiments, and interpreted model output. AM provided surface ozone data, and DT provided ozonesonde data of the three Canadian sites. KB, KS and XZ provided MAX-DOAS BrO data at Eureka, AMB

and AR provided GOME-2 tropospheric BrO columns, and XZ contributed to GOME-2 and ozonesonde data analysis. XY prepared the manuscript draft with contributions from all co-authors in both data interpretation and discussions.

**Competing interest**

The authors declare that they have no conflict of interest.

**Data availability**

The Arctic surface ozone data were retrieved from the World Data Centre for Reactive Gases (WDCRG) and archived at the NOAA Global Monitoring Laboratory (https://www.esrl.noaa.gov/gmd/ozwv/surfoz/data.html). The ozone sounding data were archived at the World Ozone and Ultraviolet Radiation Data Centre (WOUDC, http://www.woudc.org). The NOAA Air

Resources Laboratory (ARL) HYSPLIT model outputs can be accessed from their website (https://www.ready.noaa.gov/HYSPLIT.php). For GOME-2 tropospheric BrO column data, contacting corresponding authors AB and AR. For modelling outputs from UKCA and p-TOMCAT, contacting XY.

**Acknowledgements**

We thank the IASOA (www.iasoa.org) Trace Gases Working group for stimulating and encouraging this investigation. We

thank the many observers who obtained these data over many years of careful work. Ozone sounding data and surface data for Alert were provided by Environment and Climate Change Canada (ECCC). DANCEA is acknowledged for the financial support to carry out ozone measurements at Villum Research Station. The Eureka MAX-DOAS BrO measurements were made at the Polar Environment Atmospheric Research Laboratory (PEARL) by the Canadian Network for the Detection of Atmospheric Change (CANDAC), primarily supported by NSERC, CSA, and ECCC. Ozone measurements from the Tiksi

Observatory are supported by the Russian Federal Service for Hydrological and Meteorological Monitoring (Roshydromet) and the Russian Arctic and Antarctic Research Institute (AARI). The NOAA Arctic Research Program has contributed to establishing surface ozone measurement programs in Tiksi and Eureka. The Summit and Barrow surface ozone observations are collected by the NOAA Global Monitoring Laboratory. The Pallas surface ozone observations are collected by the Finnish Meteorological Institute. AB and AR gratefully acknowledge the funding by the Deutsche Forschungsgemeinschaft (DFG, German Research Foundation)-Projektnummer 268020496-TRR 172, within the Transregional Collaborative Research Center "ArctiC Amplification: Climate Relevant Atmospheric and SurfaCe Processes, and Feedback Mechanisms (AC)³" in subproject C03. The authors gratefully acknowledge the NOAA Air Resources Laboratory (ARL) for the provision of the HYSPLIT transport and dispersion model and/or READY website (http://www.ready.noaa.gov) used in this publication.

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

Table 1. Monthly mean SSA bromine depletion factor (DF) scheme applied in the NH (>45°N), which is derived from the data in the Southern Hemisphere at Cape Grim (41°S), Macquarie Island (55°S) and Dumont d'Urville (66°S) (Ayers, 1999, Legrand et al. 2016). Note that a 6-month shift of the phase is applied to match the NH seasons. A cut-off dry NaCl diameter of 10 μm is applied (i.e., DF=0 at diameter > 10 μm).

| Month | DF |
|---|---|
| January | 0.175 |
| February | 0.260 |
| March | 0.445 |
| April | 0.500 |
| May | 0.530 |
| June | 0.383 |
| July | 0.225 |
| August | 0.168 |
| September | 0.192 |
| October | 0.170 |
| November | 0.145 |
| December | 0.07 |

Table 2. Model experiments with various bromine sources from sea ice (SI), open ocean (OO) and very short-lived substances (VSLS) of bromocarbons. Two bromine depletion factor (DF) schemes are used: with seasonal cycle (Table 1) and without seasonal cycle (i.e., using a fixed DF in Table S1).

| Models and Experiments | Bromine from SI | Bromine from OO | Bromine from VSLS | DF for SSA |
|---|---|---|---|---|
| pTOMCAT_control | Yes | Yes | Yes | Seasonal |
| pTOMCAT_No_Br | No | No | No | Seasonal |
| pTOMCAT_VSLS | No | No | Yes | Seasonal |
| pTOMCAT_SI_VSLS | Yes | No | Yes | Seasonal |
| pTOMCAT_OO_VSLS | No | Yes | Yes | Seasonal |
| pTOMCAT_Fixed_DF | Yes | Yes | Yes | Fixed |
| UKCA_control | Yes | Yes | Yes | Seasonal |
| UKCA_VSLS | No | No | Yes | Seasonal |
| UKCA_SI_VSLS | Yes | No | Yes | Seasonal |
| UKCA_OO_VSLS | No | Yes | Yes | Seasonal |

Table 3: Model sensitive experiments. The parameters involved in the experiments are listed in the 2nd column. The derived sea-ice-sourced $Br_Y$ and ozone change (relative to pTOMCAT_OO_VSLS run) is in the 3rd and 5th column, respectively. The corresponding percentage (and change) of BrY and ozone are in the 4th and 6th column, respectively. The ozone difference (also relative to pTOMCAT_OO_VSLS run) is in the 5th column, with percentage of the control run result (and change) in the 6th column. The values are for April, 2007 and representing average of all the six sites.

| Experiments | Key parameters | $\Delta Br_Y$ (pptv) in April (relative to pTOMCAT_OO_VSLS) | % of pTOMCAT_control $\Delta Br_Y$ (and difference) | $\Delta O_3$ (ppbv) in April (relative to pTOMCAT_OO_VSLS) | % of pTOMCAT_control $\Delta O_3$ (and difference) |
|---|---|---|---|---|---|
| pTOMCAT_control | N=20, shape parameter α=3, scale parameter β=37.5 μm, $dm_i/dt=d_i$(*), snow salinity =3.5×Weddell Sea value, cut-off radius (dry NaCl)=l0 μm | 47.2 | 100 (0) | -20.4 | 100 (0) |
| pTOMCAT_spectrum_1 | same as pTOMCAT_control but N=10 | 37.8 | 79.9 (-20.1) | -17.0 | 83.5 (+16.5) |
| pTOMCAT_spectrum_2 | same as pTOMCAT_control but N=1, α=2, β=70 μm and $dm_i/dt$=constant | 27.3 | 57.8 (-42.2) | -12.5 | 61.1 (+38.6) |
| pTOMCAT_low_salinity | same as pTOMCAT_control but snow salinity =1× Weddell Sea value | 18.9 | 40.0 (-60.0) | -8.0 | 39.0 (+61.0) |
| pTOMCAT_high_salinity | same as pTOMCAT_control but snow salinity =10× Weddell Sea value | 92.0 | 194.8 (+94.8) | -28.1 | 137.5 (-37.5) |
| pTOMCAT_2×DF | same as pTOMCAT_control but 2×DF(**) | 87.1 | 184.4 (+84.4) | -27.7 | 135.7 (-35.7) |
| pTOMCAT_0.5×DF | same as pTOMCAT_control but 0.5×DF | 24.9 | 52.7 (-47.3) | -11.1 | 54.2 (+45.8) |
| pTOMCAT_SSA20μm | same as pTOMCAT_control run but cut-off radius=20 μm | 67.3 | 142.3 (+42.3) | -24.7 | 121.1 (-21.1) |
| pTOMCAT_SSA5μm | same as pTOMCAT_control but cut-off radius=5 μm | 21.2 | 44.9 (-55.1) | -9.1 | 44.6 (+45.4) |

*: $m_i$ is water mass of the particle, and $d_i$ is diameter.

**: if the 2×DF value is > 1.0, then a maximum value of 1.0 is used.

Table 4. Correlation coefficients ($R$) at each site between various variables used in Fig. 6 and 7. Note that [*] indicates probability value < 0.1, [**] <0.01 and [***] < 0.001.

| $R$ | Obs $O_3$ vs model surface $O_3$ | Obs $O_3$ vs model $BrO_{surface}$ | Obs $O_3$ vs model $BrO_{trop}$ | Obs $O_3$ vs GOME-2 $BrO_{trop}$ | Model surface $O_3$ vs model $BrO_{surface}$ | Model $BrO_{trop}$ vs GOME-2 $BrO_{trop}$ | Model $BrO_{trop}$ vs model $BrO_{surface}$ |
|---|---|---|---|---|---|---|---|
| Ticksi | 0.68[***] | -0.49[***] | -0.53[***] | -0.62[***] | -0.78[***] | 0.59[***] | 0.96[***] |
| Barrow | 0.49[***] | -0.04 | -0.23[***] | -0.32[**] | -0.42[***] | 0.37[***] | 0.87[***] |
| Summit | 0.63[***] | -0.19[***] | 0.03 | 0.09 | 0.00 | 0.16[*] | 0.42[***] |
| Villum | 0.76[***] | -0.51[***] | -0.34[***] | -0.42[***] | -0.50[***] | 0.33[**] | 0.57[***] |
| Alert | 0.24[***] | -0.05[*] | -0.22[***] | 0.03 | -0.36[***] | 0.30[**] | 0.59[***] |

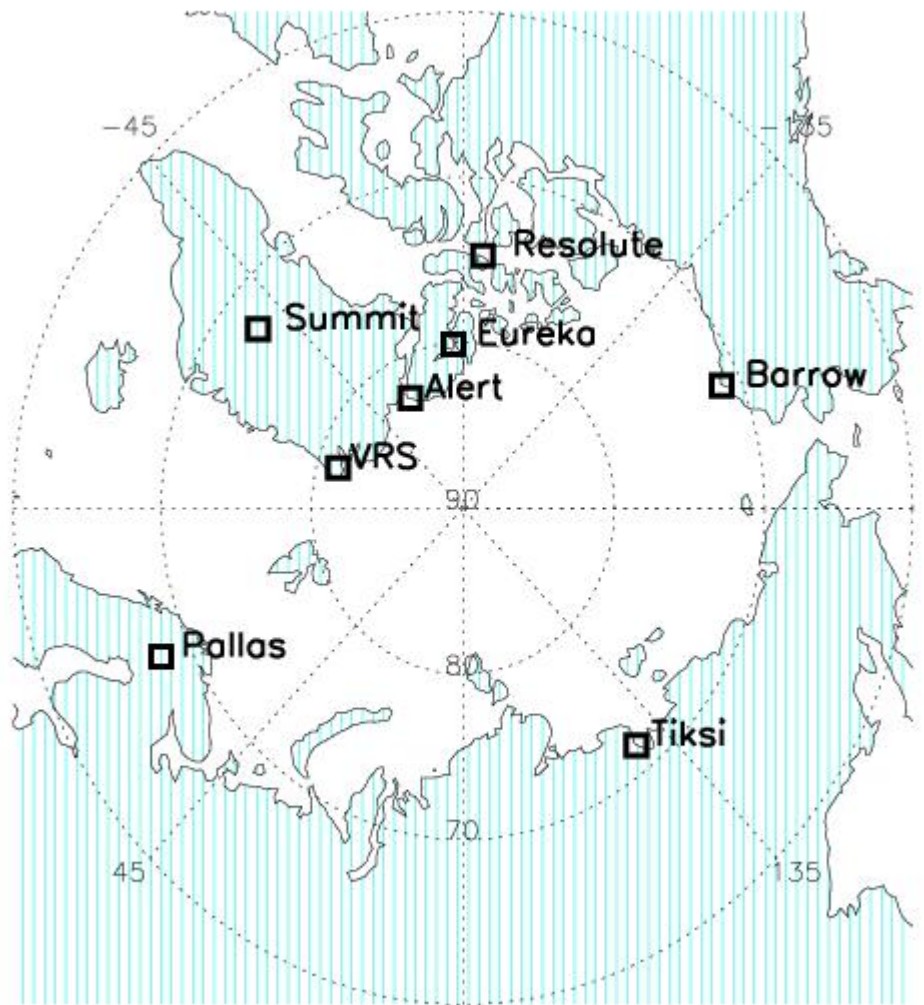

Figure1: Map of the Arctic showing the locations of the eight Arctic sites where either surface ozone and/or ozonesonde data are used in this study.

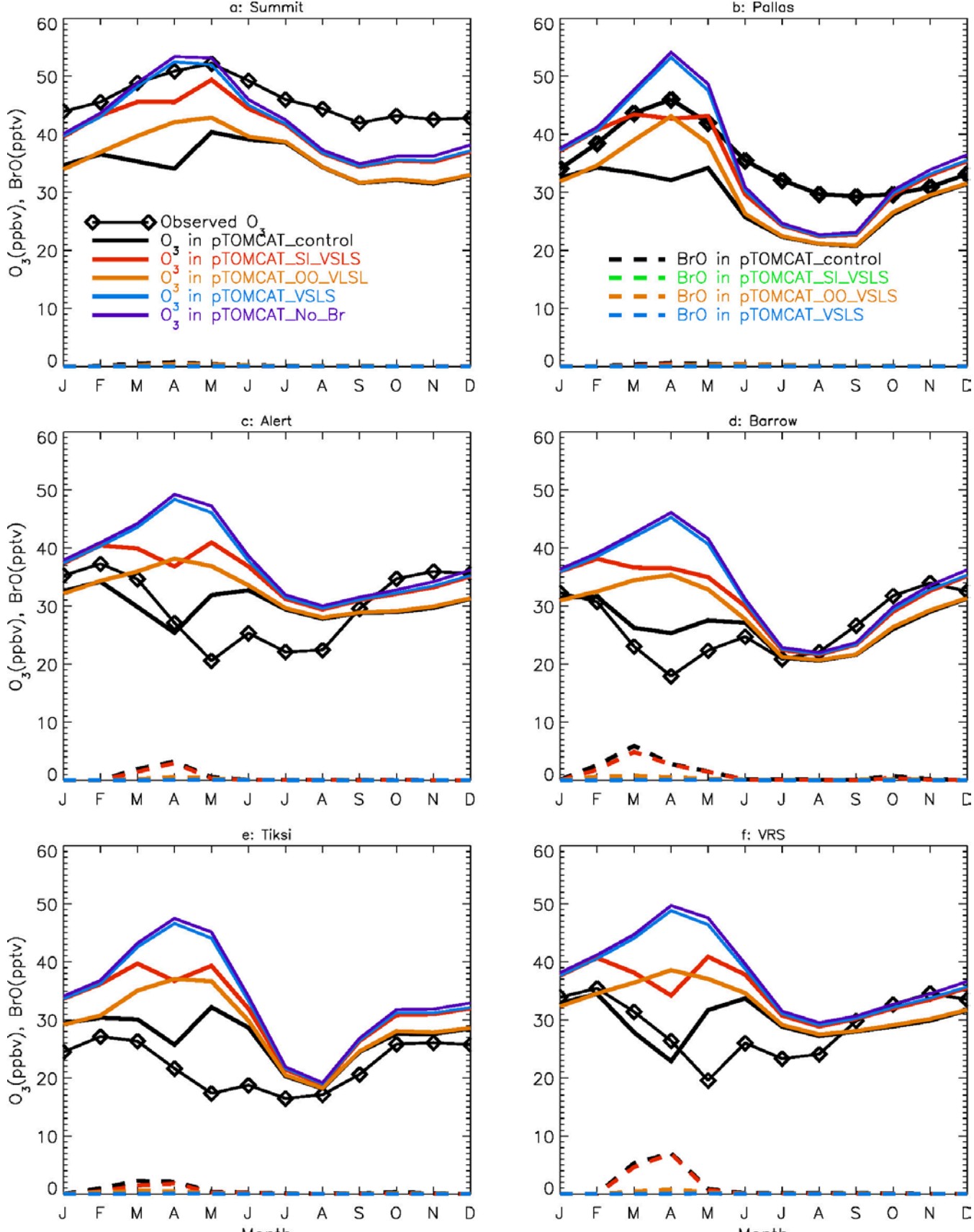

Figure 2: Climatology of monthly mean surface ozone (solid black line with diamond symbols) at six Arctic sites. The observed data are the average of 2000-2016 at Summit, 1995-2012 at Pallas, 1992-2012 at Alert, 1974-2016 at Barrow, 2011-2016 at Tiksi and 1980-2014 at VRS Research Station. Model surface ozone concentrations (VMRs) from various experiments are shown in colourful solid lines with BrO in dashed lines based on an integration of 2006-2008.

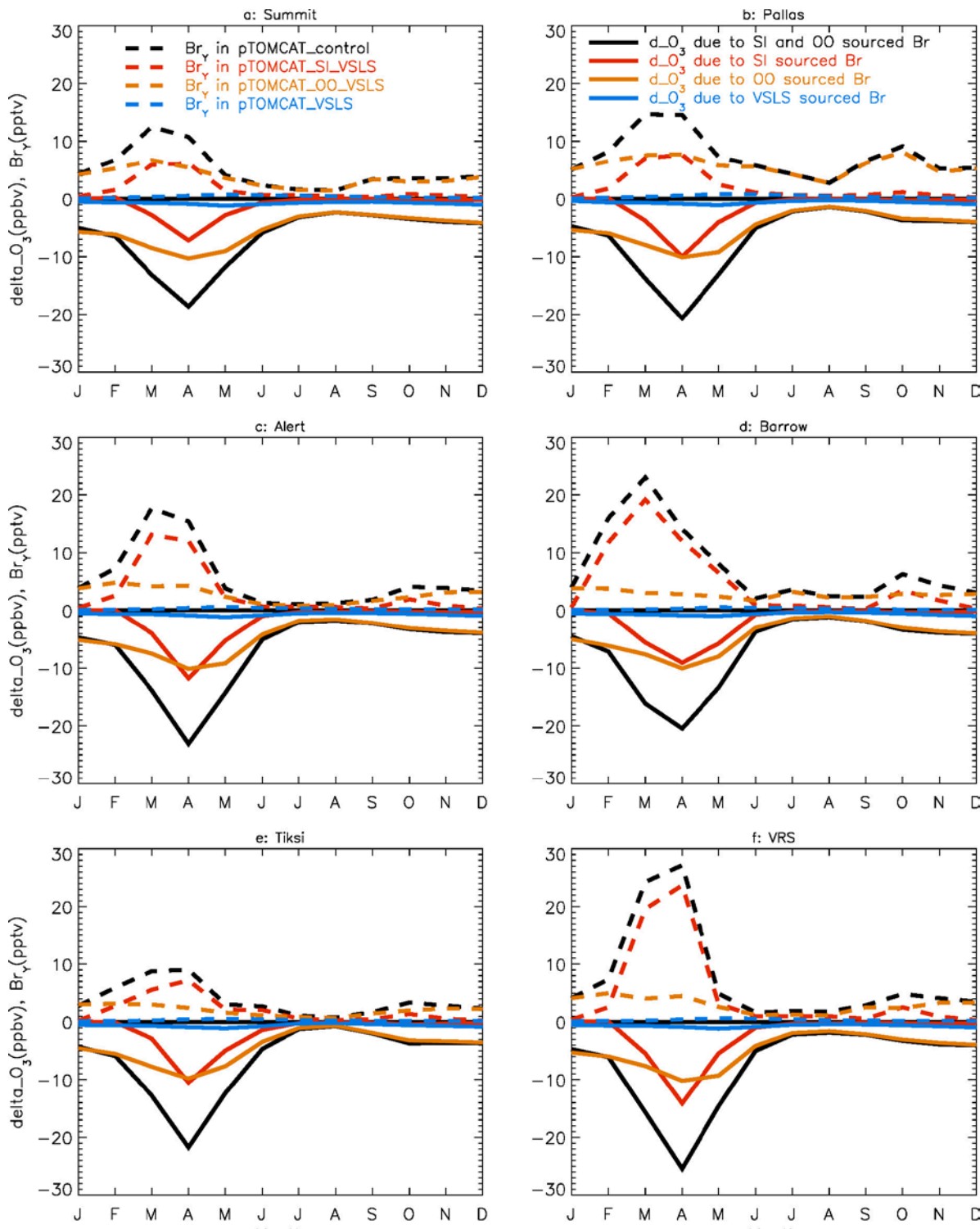

Figure 3: Ozone changes in response to alteration of various bromine sources. E.g., solid black line =pTOMCAT_control - pTOMCAT_VSLS representing both SI and OO contributions, solid red line =pTOMCAT_control - pTOMCAT_OO_VSLS representing SI contribution only, solid orange line =pTOMCAT_control - pTOMCAT_SI_VSLS representing OO contribution only, and solid blue line =pTOMCAT_VSLS - pTOMCAT_No_Br representing VSLS contribution only. Dashed lines represent total inorganic bromine (Br$_Y$) in various model runs: the dashed black line is the pTOMCAT_control run representing all bromine source contributions, the dashed red line is the pTOMCAT_SI_VSLS run representing SI and VSLS contributions, the dashed orange line is the pTOMCAT_OO_VLSL run representing OO and VSLS contributions, and the dashed blue line is the pTOMCAT_VSLS run representing VSLS contribution only.

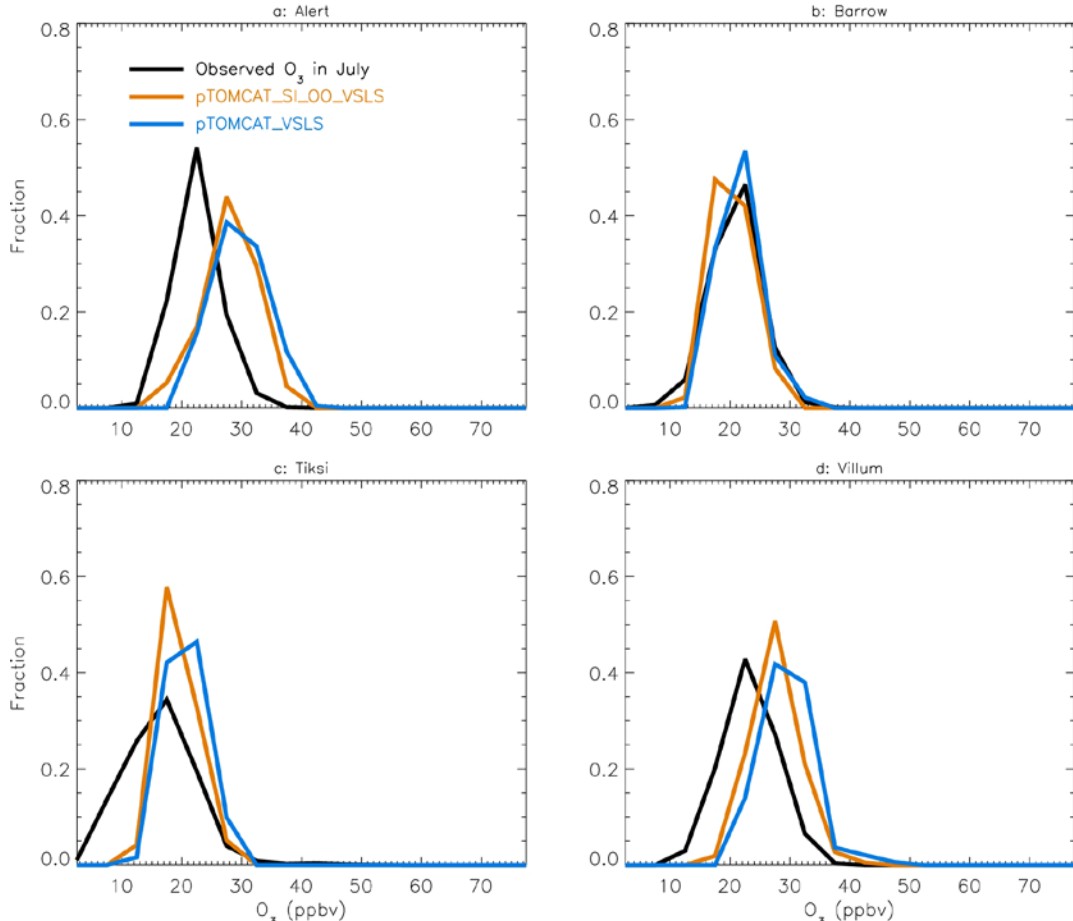

Figure 4: July surface ozone fractional distribution (with a bin interval of 5 ppbv). The observed fractional distribution is shown in black line, with pTOMCAT_control result shown in solid orange and pTOMCAT_VSLS shown in solid blue.

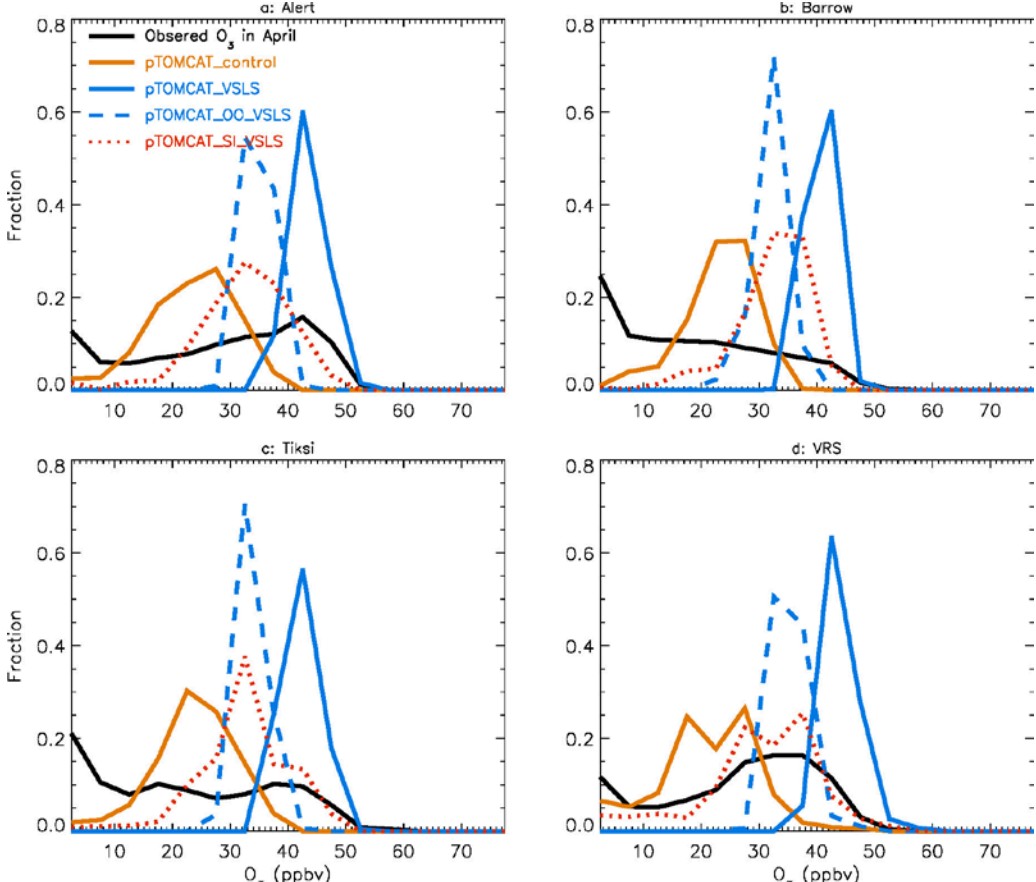

Figure 5: Same as Fig. 4 but for April. Also shown are the pTOMCAT_OO_VSLS results using dashed blue line and the pTOMCAT_SI_VSLS result in dashed orange.

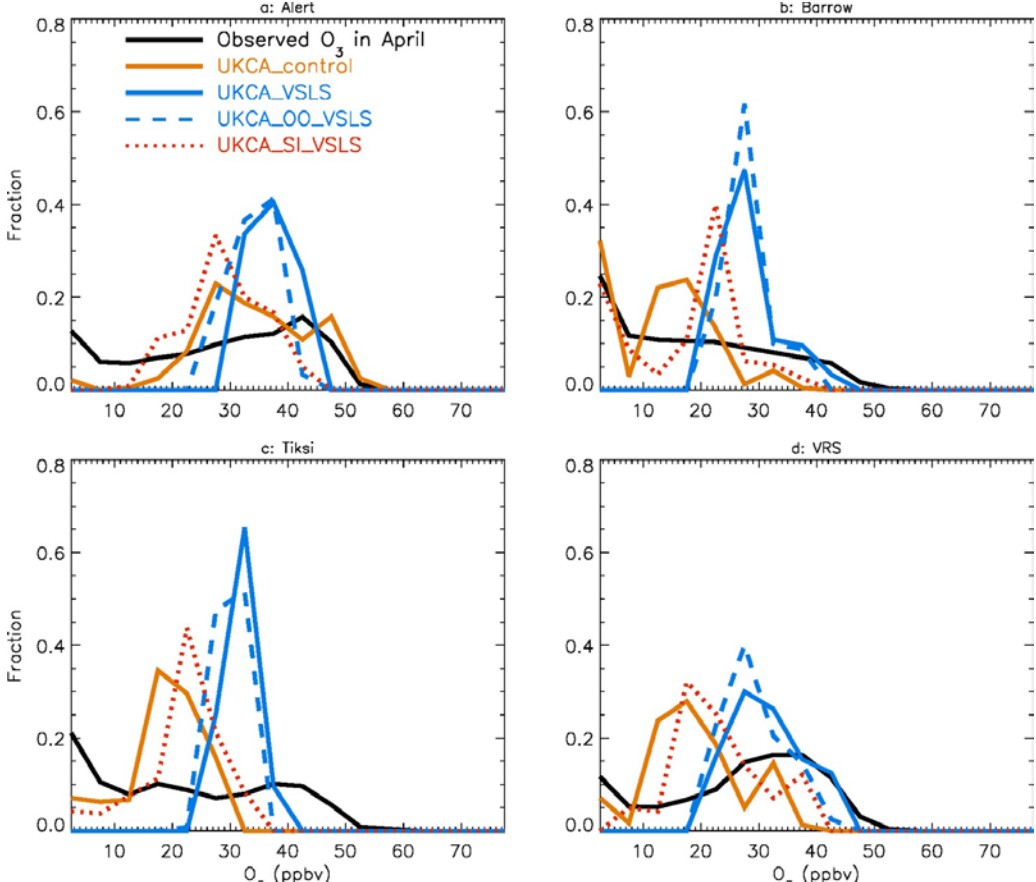

Figure 6: Same as Fig. 5 but for UKCA model results.

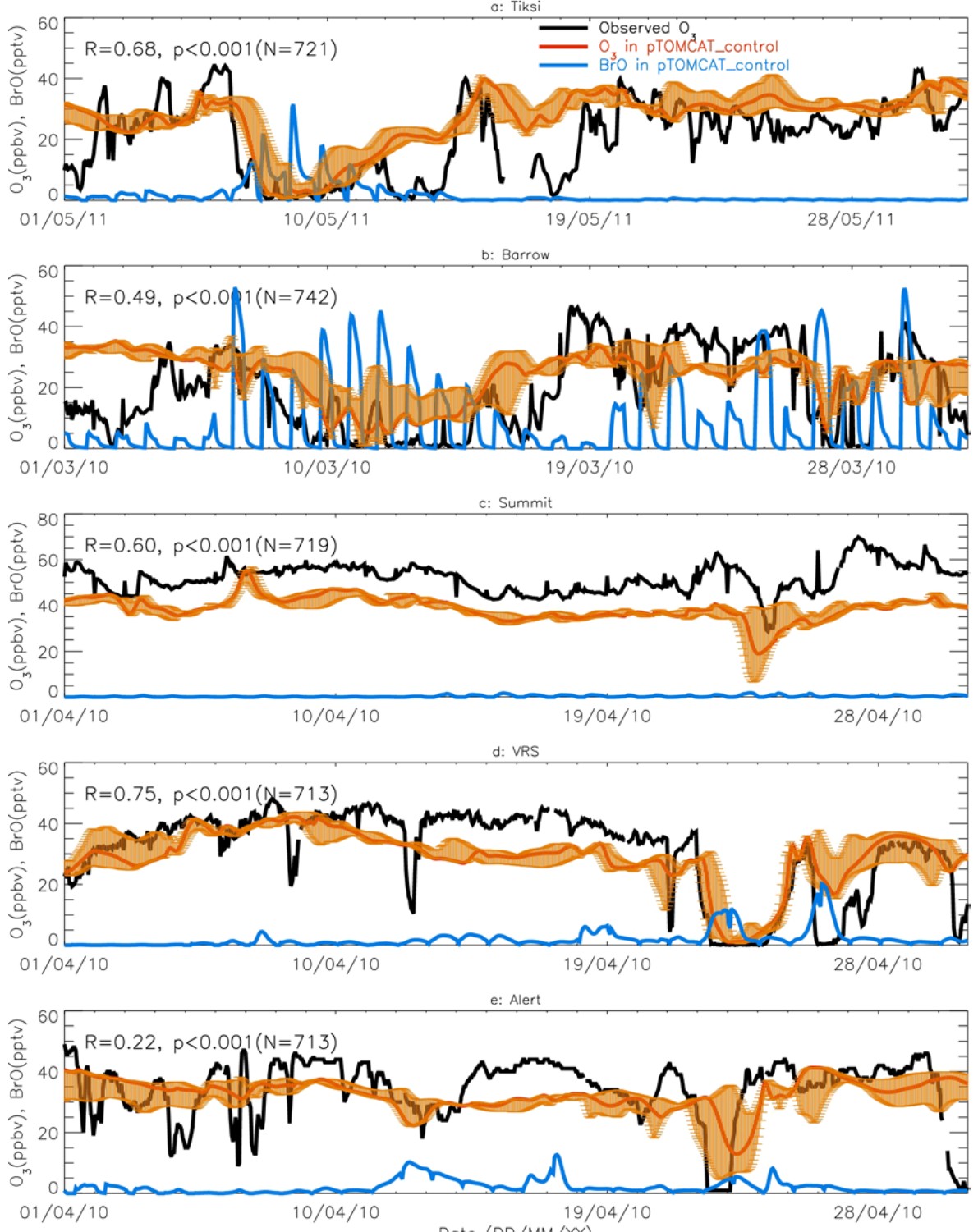

Figure 7: 1-month long time-series of surface ozone and BrO at (a) Tiksi (May 201), (b) Barrow (March 2010), (c) Summit (April 2010), (d) Villum (April 2010) and (e) Alert (April 2010). Observed ozone is shown in black, with pTOMCAT_control ozone shown in bold red representing value of the nearest central gridbox to the observation site. The bold blue line is central gridbox BrO. Note that maximum and minimum ozone, which are shown in orange bars, are taken from the five adjacent gridboxes next to the central gridbox to highlight the possible range of the tracer concentrations.

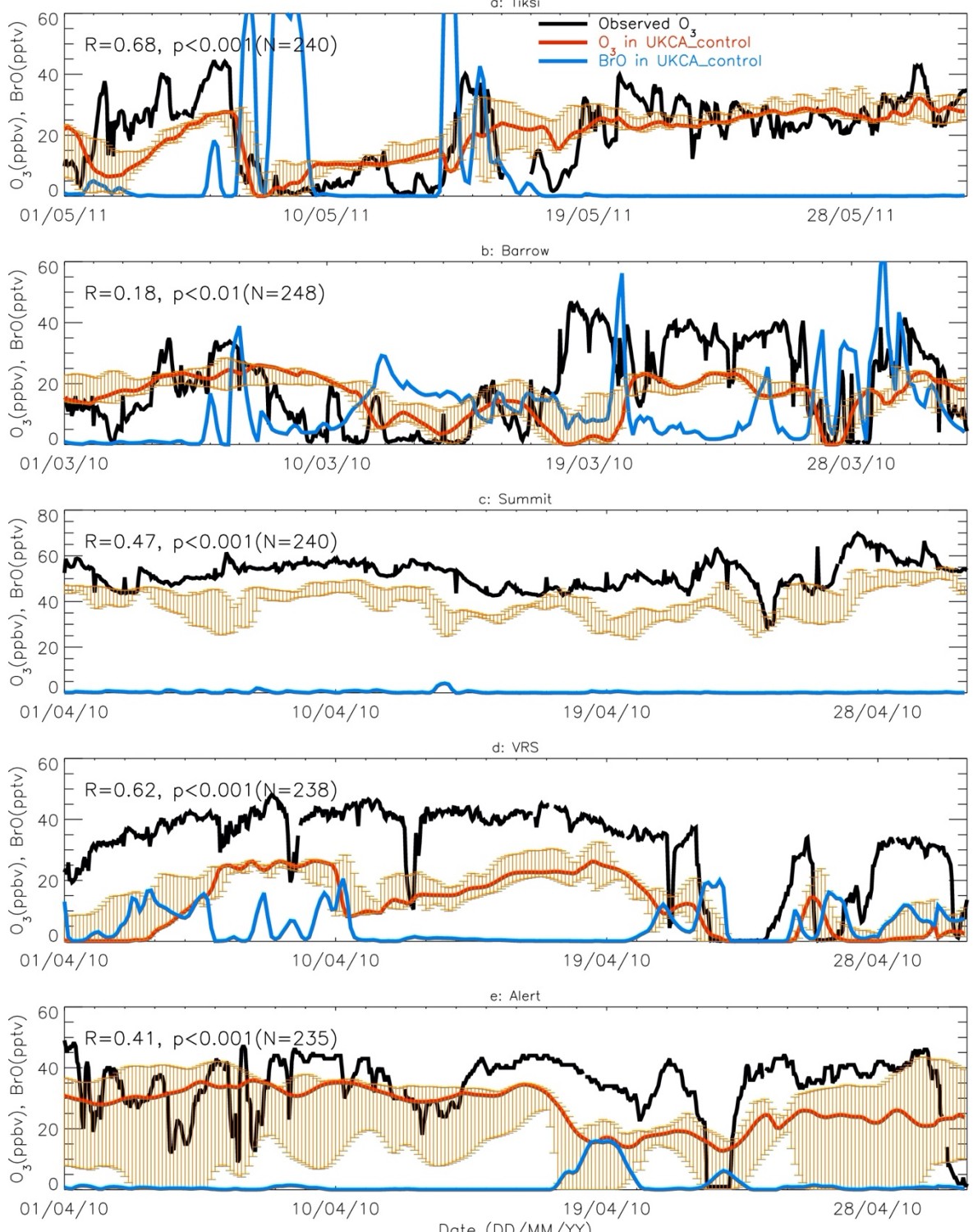

Figure 8: Same as Fig. 7 but for UKCA_control run results.

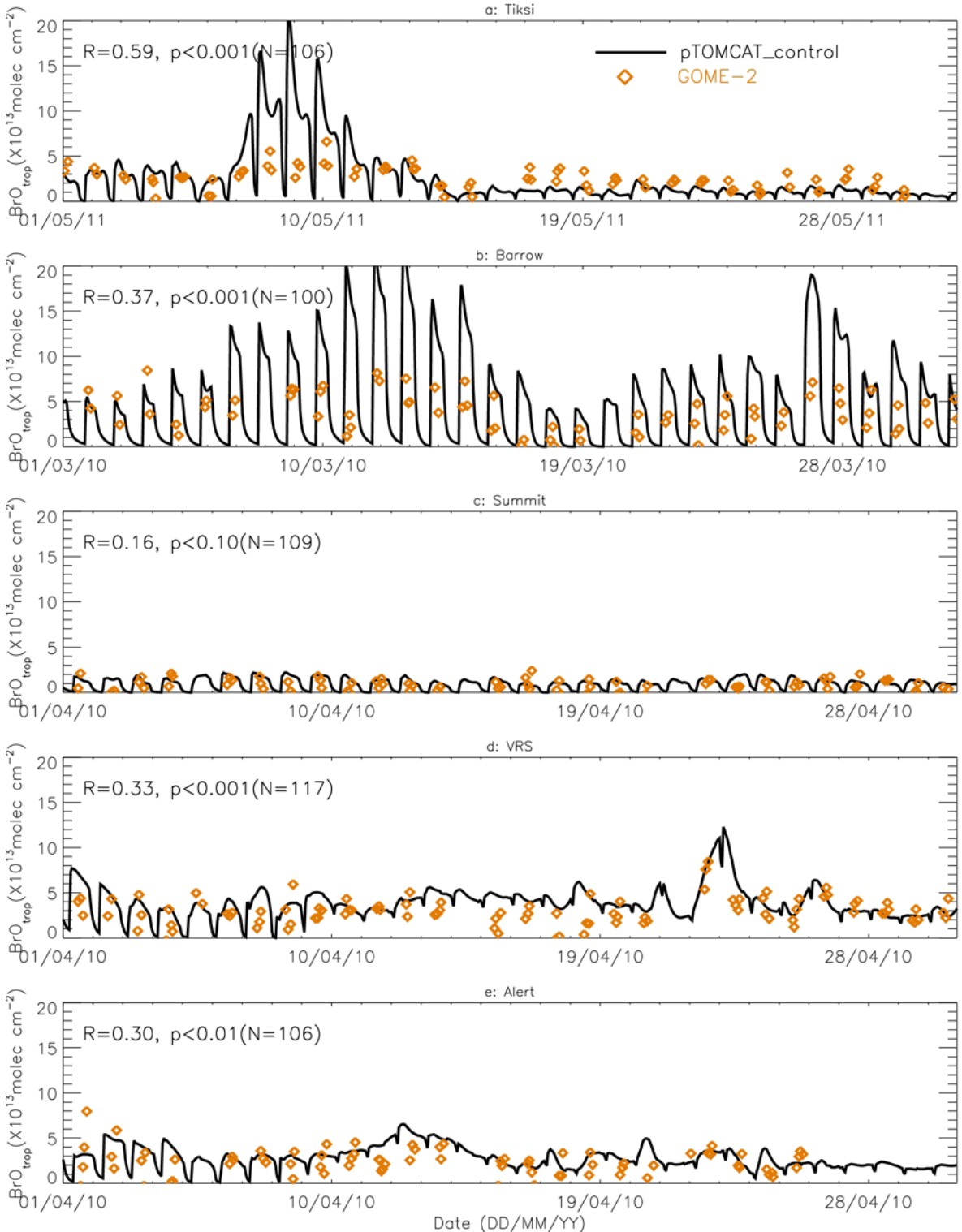

Figure 9: Time-series of tropospheric BrO column from GOME-2 (orange diamond symbols) and pTOMCAT_control run BrO (black line). The correlation coefficient *R* and statistics significance level p at each site is given.

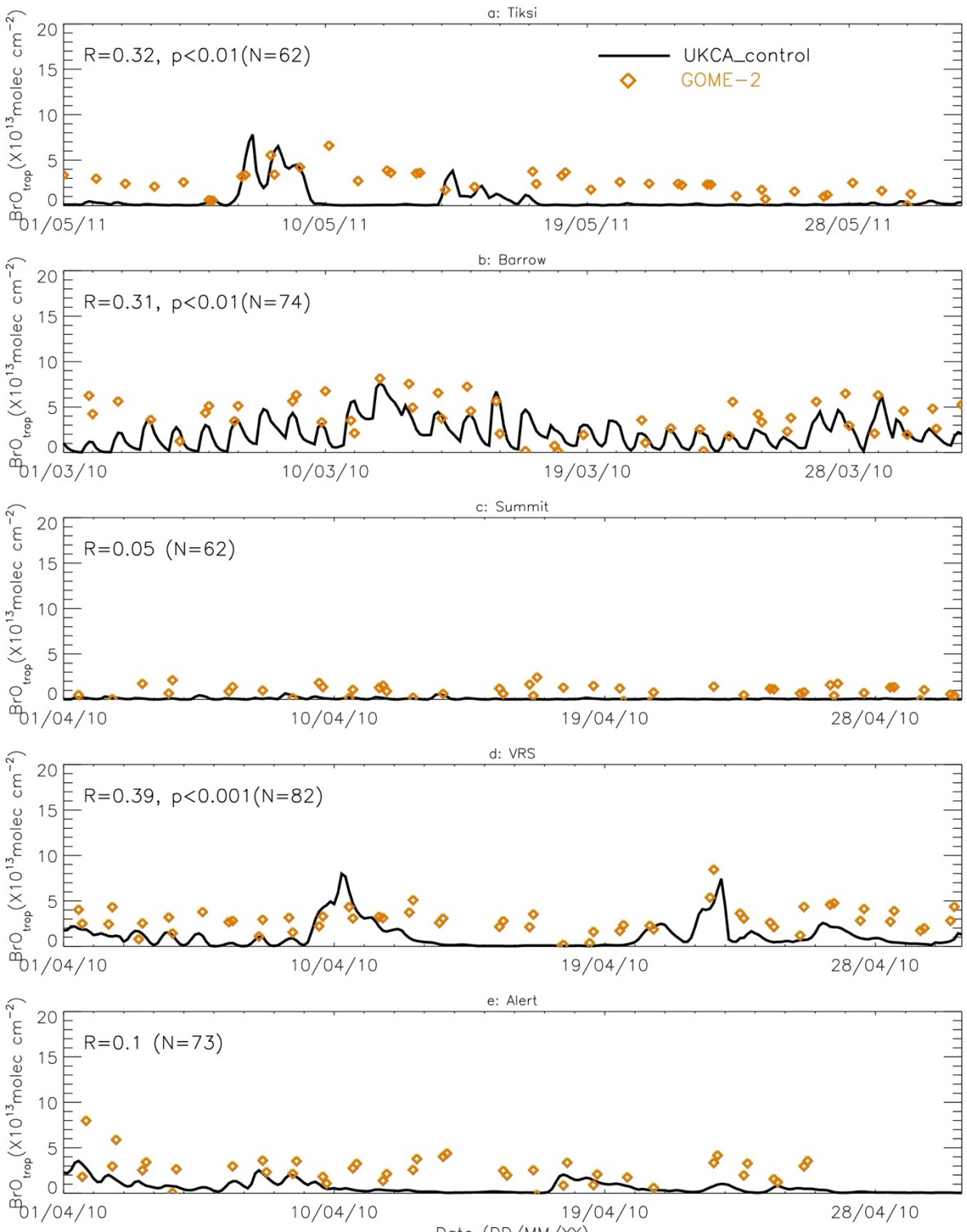

Fig. 10: Same as Fig. 9 but for UKCA_control run results.

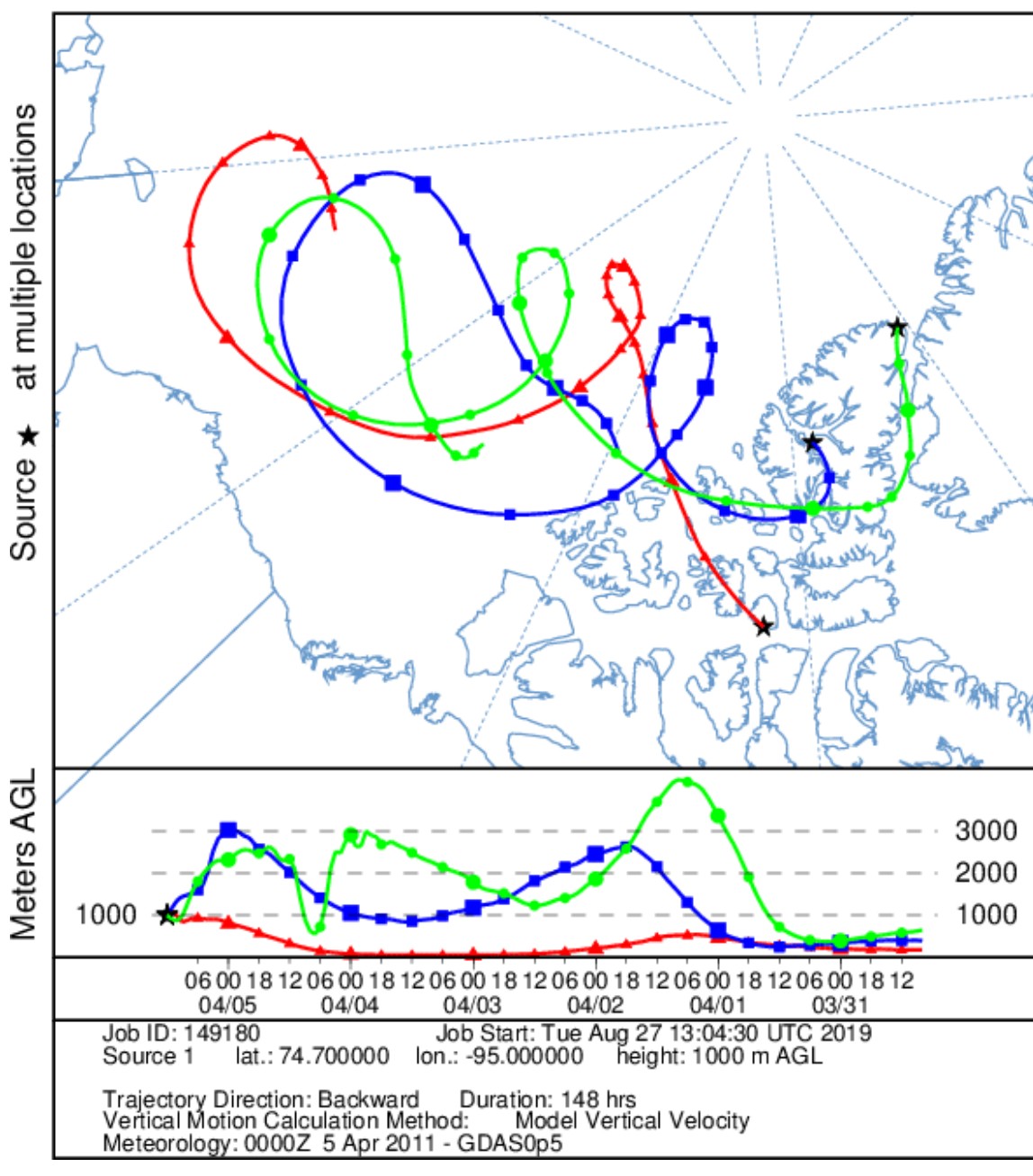

Figure 11: HYSPLIT 6-day back trajectories for Resolute, Eureka and Alert ending at 12:00 UTC on 5 April 2011.

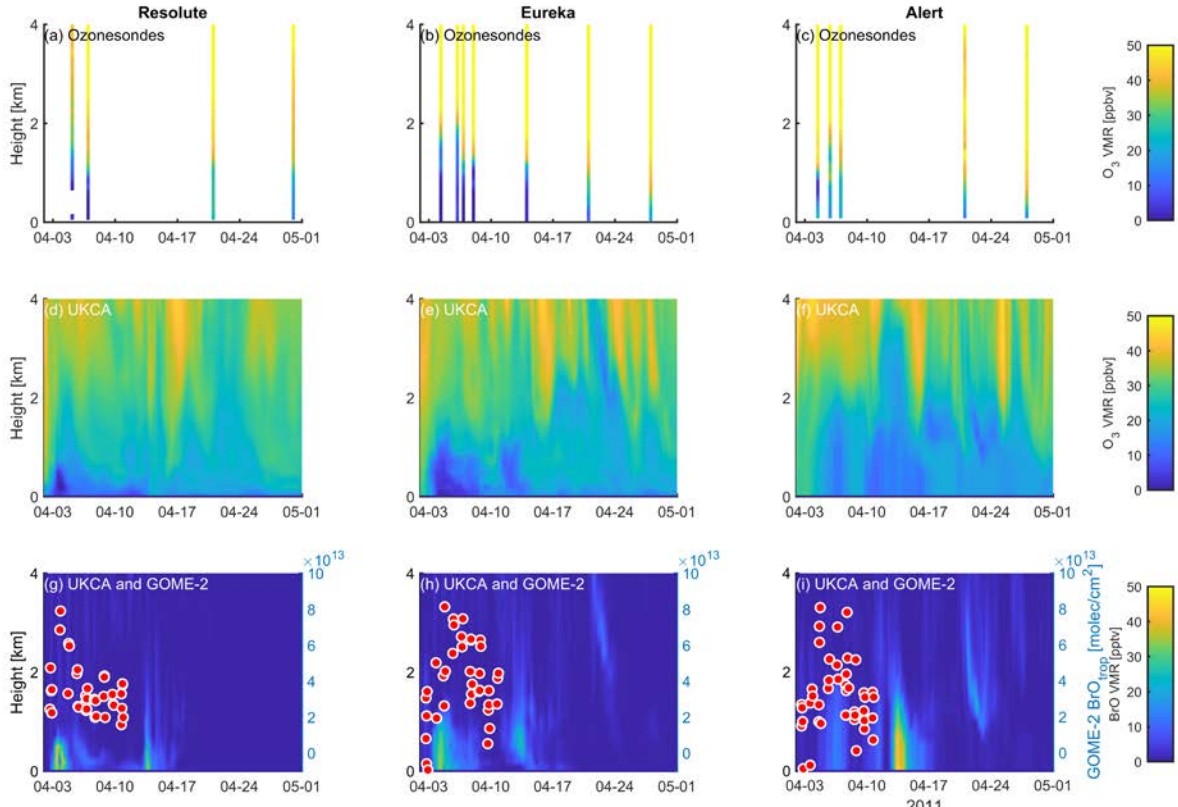

Figure 12: Profile of ozonesonde (0-4km) at Resolute (a), Eureka (b), and Alert (c) during April 2011. UKCA_control ozone profiles (d-f) and BrO profiles (g-i) are also plotted. GOME-2 overpass data (tropospheric column BrO) of the period 1-10 April 2011 is also plotted in g-i, which is detailed in Fig. 10.

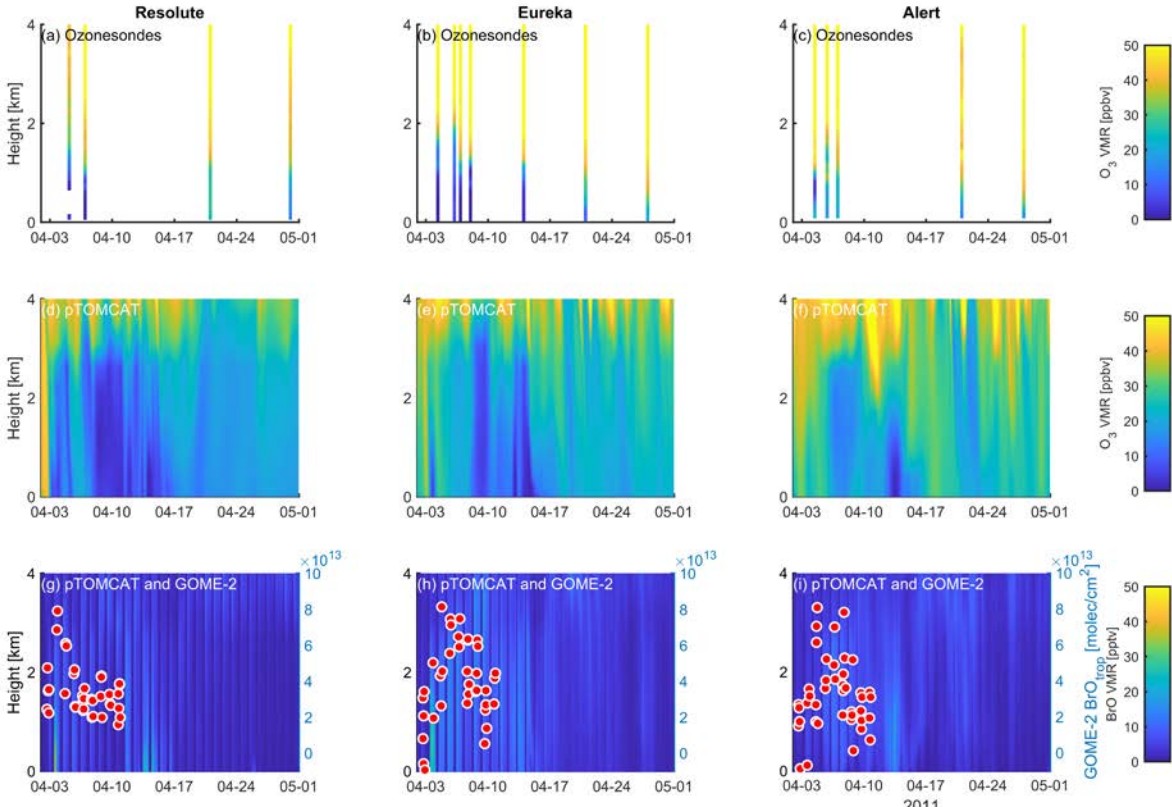

Figure 13: Same as Fig.12 but for pTOMCAT_control run results.

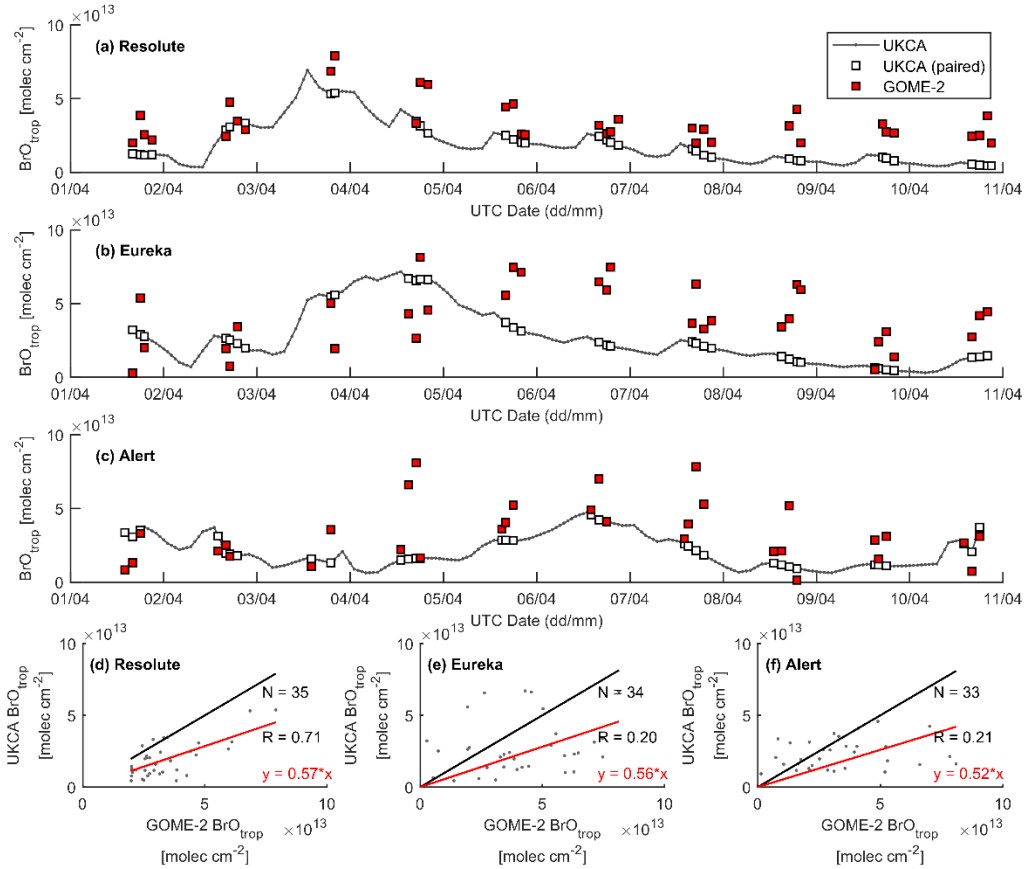

Figure 14: Time series of GOME-2 and UKCA_control tropospheric BrO column at (a) Resolute, (b) Eureka, and (c) Alert for the period 1-10 April 2011. Correlation plots between the model and GOME-2 are shown in d-f, with black line representing 1:1 plot and red line representing regression fit.

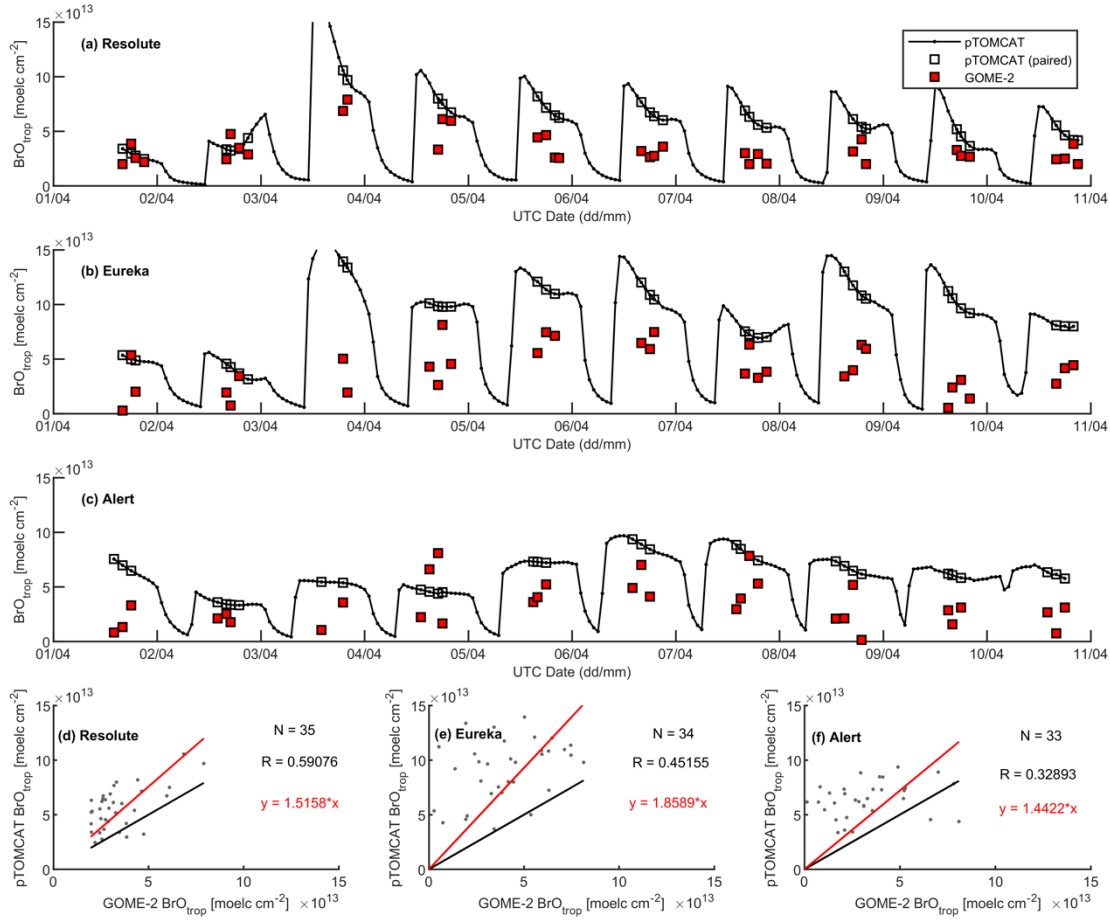

Figure 15: Same as Fig. 14 but for pTOMCAT_control run results.

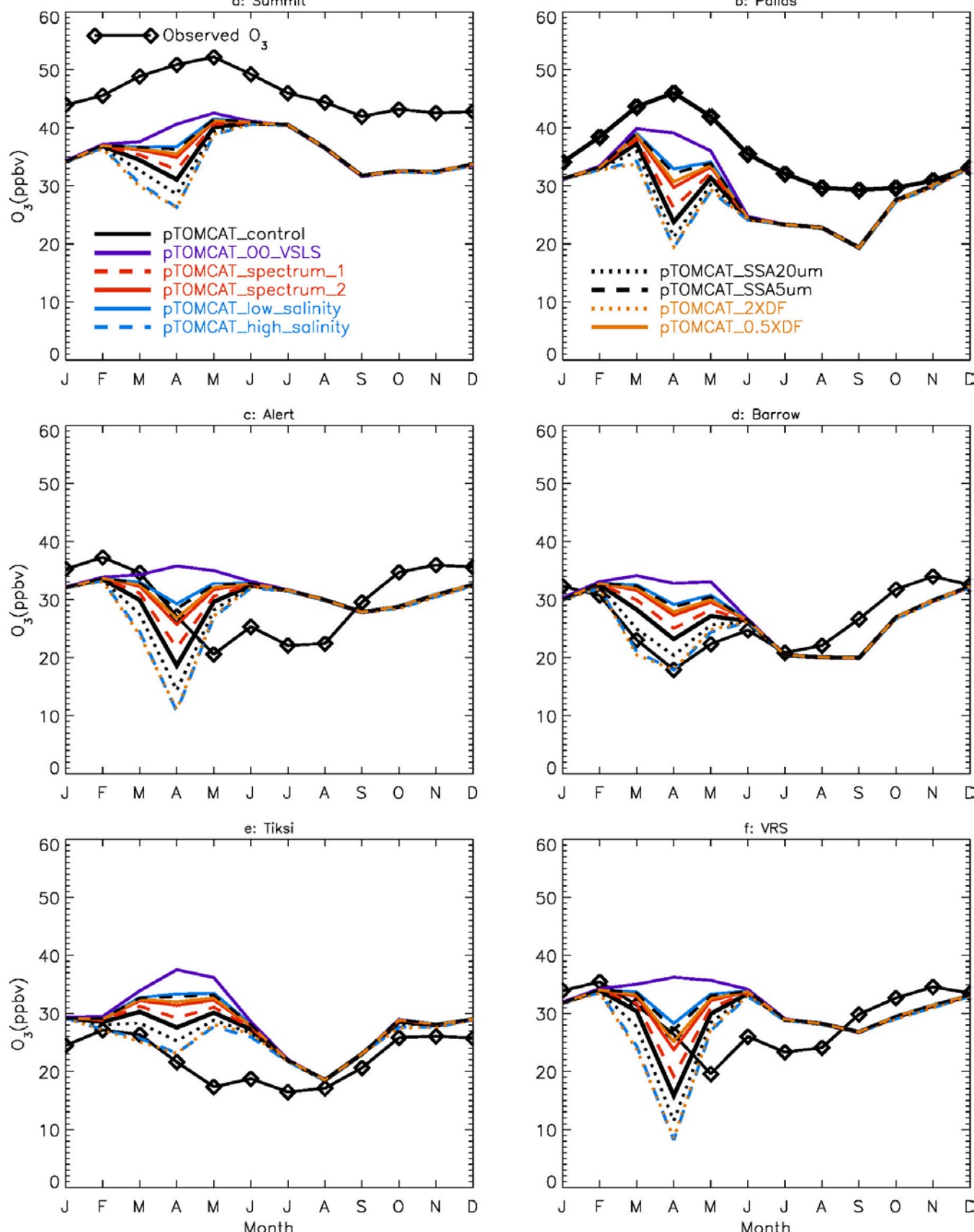

Fig. 16: Monthly mean ozone at the six sites in the Arctic. Ozone observations are climatology and simulated ozone outputs are only for year 2007.

