# Peer review of "Pan-Arctic surface ozone: modelling vs measurements"

_Atmospheric Chemistry and Physics, 2019_

## Referee Comment (RC1) · Anonymous Referee #2 · 8 May 2020

This study uses a suite of surface observations, ozone sonde measurements and satellite observations to evaluate pan-Arctic surface ozone in two global models (a chemical transport model (CTM) and a coupled chemistry-climate model) and the impacts of halogen chemistry on simulated seasonal cycles and modelled ozone during springtime ozone depletion events. The study serves as a useful benchmark for demonstrating the importance of inclusion of bromide chemistry in models for the simulation of Arctic surface ozone, and when and where this might lead to model improvement (as well as degradation). The assessment is novel, and it is a nice synthesis of surface data, satellite data and modelling. The paper is well written with clear figures. In general, the paper is worthy of publication in ACP, but I would recommend that the following points are addressed before acceptance.

[Figure]

A weak aspect is the exploitation of the two different models. It seems a lot of effort to run both models for the analysis, yet not very much is made of comparing their performance and discussing possible reasons for their different performance or what can be learned from this. In many sections and figures, only one of the two models is shown (different single models for different sections), which is not well justified.

Specific points:

Page 3, paragraph 2: There have in fact been multi-model assessments of Arctic surface ozone in global CTMs. See Monks et al., (2015), Emmons et al., (2015), and an older study by Shindell et al., (2008). Monks et al and Shindell et al both show overprediction of surface ozone at Barrow in spring, likely as a result of missing halogen chemistry. However, Emmons et al show a general model *underprediction* over the depth of the Arctic troposphere in April compared with ozone sondes, suggesting that the halogen-induced bias may not be pervasive in the Arctic troposphere. It would be helpful to see these previous studies highlighted in the text for context.

Page 5, line 3: "The retrievals were performed on a 0-4 km grid with 0.2 km resolution." Not clear what this means. What is a 0-4km grid?

Page 5, line7: Both models are driven by ERA-Interim data. For UKCA, please briefly explain what this means for the climate model. i.e. does this imply nudging with a certain degree of relaxation? Over what altitude range? It is important to recognise that this is different from a purely offline model (such as pTOMCAT). What else is prescribed / free-running between the models? Clouds? Surface fluxes?

Page 5, line 23: Is the Law et al., (2000) study the most up-to-date reference for the model chemistry scheme? How up-to-date is the kinetic data used? How do these data compare with that used in the UKCA model for the same tropospheric reactions? Does p-TOMCAT include non-halogen related heterogeneous chemistry (it seems that UKCA does)- e.g. $N_2O_5$ hydrolysis on aerosol, which is likely important for winter / early spring ozone and $NO_y$ in the Arctic. Given the focus on comparing ozone performance

between the models, it is important to acknowledge any important differences in the chemical schemes of the models.

Page 7, line 15: Care needs to be taken in over-interpreting the reason for differences between the models and assigning this to mainly physical parameters (and I agree that such differences between a climate model and CTM would be expected - although see also my comment on nudging), as it may be that there are important differences between the model chemical schemes (see point above). This is why it would be useful to point out more about these potential differences. I am not sure it is the case that only or a dominance of physical factors can be assumed.

Page 7, line 32: Effect of dry deposition on ozone abundances during long-range transport into the Arctic. A key references here showing suppressed high latitude ozone due to deposition loss to vegetation in Siberia is Stjernberg et al., (2012).

Section 4.1: Discussion of model and observed seasonal cycles. I think it is worth explicitly pointing out that inclusion of the halogen chemistry the control pTOMCAT model leads to severe under-estimation of spring ozone at Summit and Pallas.

It does not seem obvious to me why in presentation of the results in Section 4.3.1 model output switches from using p-TOMCAT to UKCA. Perhaps I have missed something in the applicability of the simulations to different periods. In general, only one model is shown for each part of the results comparing with observations. Would it be more informative to show both models where possible?

Throughout: I find the use of the labels "pTOMCAT_SI_OO_VSLS" and "UKCA_SI_OO_VSLS" to name the two control runs overly complicated and distracting when reading. It is better just to call these "pTOMCAT_control" and "UKCA_control" in the text. The names of the other experiments are then enough to highlight what is missing/included for the other runs.

I would recommend a more explicit short "Summary" or "Conclusions" section to definitively set out the key findings of the study and their context in the wider picture. At the moment, the final paragraph is a bit brief and needs to be separated from the main discussion.

Figure 2 caption: "in various experiments" - please clarify in the caption from which model.

Figure 6 caption, please correct date "(May 201)"

References

Emmons, L. K., Arnold, S. R., Monks, S. A., Huijnen, V., Tilmes, S., Law, K. S., Thomas, J. L., Raut, J.-C., Bouarar, I., Turquety, S., Long, Y., Duncan, B., Steenrod, S., Strode, S., Flemming, J., Mao, J., Langner, J., Thompson, A. M., Tarasick, D., Apel, E. C., Blake, D. R., Cohen, R. C., Dibb, J., Diskin, G. S., Fried, A., Hall, S. R., Huey, L. G., Weinheimer, A. J., Wisthaler, A., Mikoviny, T., Nowak, J., Peischl, J., Roberts, J. M., Ryerson, T., Warneke, C., and Helmig, D.: The POLARCAT Model Intercomparison Project (POLMIP): overview and evaluation with observations, Atmos. Chem. Phys., 15, 6721–6744, https://doi.org/10.5194/acp-15-6721-2015, 2015.

Monks, S. A., Arnold, S. R., Emmons, L. K., Law, K. S., Turquety, S., Duncan, B. N., Flemming, J., Huijnen, V., Tilmes, S., Langner, J., Mao, J., Long, Y., Thomas, J. L., Steenrod, S. D., Raut, J. C., Wilson, C., Chipperfield, M. P., Diskin, G. S., Weinheimer, A., Schlager, H., and Ancellet, G.: Multi-model study of chemical and physical controls on transport of anthropogenic and biomass burning pollution to the Arctic, Atmos. Chem. Phys., 15, 3575–3603, https://doi.org/10.5194/acp-15-3575-2015, 2015.

Shindell, D. T., Chin, M., Dentener, F., Doherty, R. M., Faluvegi, G., Fiore, A. M., Hess, P., Koch, D. M., MacKenzie, I. A., Sanderson, M. G., Schultz, M. G., Schulz, M., Stevenson, D. S., Teich, H., Textor, C., Wild, O., Bergmann, D. J., Bey, I., Bian, H., Cuvelier, C., Duncan, B. N., Folberth, G., Horowitz, L. W., Jonson, J., Kaminski, J. W., Marmer, E., Park, R., Pringle, K. J., Schroeder, S., Szopa, S., Takemura, T., Zeng, G., Keating, T.

J., and Zuber, A.: A multi-model assessment of pollution transport to the Arctic, Atmos. Chem. Phys., 8, 5353–5372, https://doi.org/10.5194/acp-8-5353-2008, 2008.

Stjernberg, A-C. ,Skorokhod, A. & Paris, J-D. & Elansky, N., Nedelec, P. & Stohl, Andreas, Low concentrations of near-surface ozone in Siberia. Tellus B. 64. 10.3402/tellusb.v64i0.11607, 2012.

---

## Referee Comment (RC2) · Anonymous Referee #1 · 11 May 2020

The authors present results of model simulations of halogen chemistry including Arctic ozone and bromine monoxide (BrO) abundance. These simulation results are compared to observations of the same species. Two different models are used, p-TOMCAT and UKCA, and are compared. A critical part of the modeling effort is the need to model the production of sea-salt derived aerosol particles (SSA), which is done here by including both modeled open ocean (OO) and sea ice (SI) processes based upon a blowing snow model. Overall, I think that the manuscript is not carefully argued and would need major revisions to be considerable for publication in ACP. I come to this decision based upon the following major points. After discussion of the major issues, I note a number of smaller or typographical issues.

A major lack of the manuscript comes from the belief that the process trying to be mod-

eled is uniquely represented by the model. This belief underlies statements such as in the abstract, where the manuscript states "...reproducing both ODEs and BEEs in the Arctic indicates that the relevant parameterizations implemented in the models work reasonably well, which supports the proposed mechanism of SSA and bromine production from blowing snow on sea ice." It is true that inclusion of this modeled process reduces springtime ozone and thus getting the model results closer to observations. However, to really make this statement (and many others like it in the manuscript), other models of sea-ice-related reactive bromine release would have need to have been considered and shown to be of lower skill than the proposed blowing snow SI-sourced bromine model. Without considering if other sea-ice-related processes could work effectively, the authors have not shown that this process is uniquely the one that is responsible. Other models in the literature could release of reactive bromine from snowpack (e.g. Pratt et al. 2013), production of SSA from wind over open water in sea ice leads between ice floes (e.g. Kirpes et al., 2019). To be devil's advocate, one could make a model of seasonally varying dry deposition of ozone to snowpack, which could be tuned to get agreement with the observed average monthly ozone data. Should the agreement of this model with average ozone observations then be taken as a sign that this process is the actual physical process that is occurring? The authors neither show that their model is unique nor do they present external validation of aspects of their model. The use of Antarctic snow salinity data scaled by what appear to be a number of tuning parameters without Arctic validation does not give confidence in this SSA production model. Specifically, on page 5, line 37, the authors indicate that they have altered parameters of their model. Why? To what end? Why did they change the snow salinity by 3.5 times Antarctic data? The authors indicate on page 10, line 37 that "We lack snow data on Arctic sea ice to constrain the dataset used in this study". There are papers on snow on sea ice in the Arctic (e.g. Pratt et al. 2013, Krnavek et al., 2012, Xu et al., 2016). How do these observations compare with their scaled salinity. The use of these tunings without any justification of what is trying to be obtained gives me caution in believing statements like the one at the start of this paragraph, and particularly with

attribution of ozone losses to processes, such as the abstract, which states: "In April, bromine chemistry can cause a net loss (monthly mean) of 10-20 ppb, with almost half attributable to open-ocean-sourced bromine and the rest to sea-ice-sourced bromine". I find the link between the modeled processes and real processes, which underlies this statement, suspect given the lack of uniqueness and I find that attribution questionable given the apparent tuning of processes to some unknown end.

It is not clear from this study how reactive bromine is produced from the SSA bromide. The section on this topic is 3.1, and also refers to the supplemental table 1. First, it is not clear what the model is doing. Is the model actually considering heterogeneous chemical reactions that would convert SSA bromide (Br-) to gas-phase reactive bromine precursors (e.g. Br2)? I expect that the model is not actually considering these reactions, but instead simply multiplying the DF * SSA to get "lost" Br- from the SSA and making this into Br2. The manuscript should be clear as to what is actually being modeled. If the process is simply taking the bulk DF times SSA bromide, then there are a number of physical and chemical problems with this approach. For example, we know that reactive bromine deactiviates fairly rapidly (e.g. reaction of Br with H2CO to form HBr), which then partitions to the aerosol particle phase, increasing Br- in the SSA (and thus reducing the DF). This recycling of reactive bromine is needed for persistent (e.g. multi-day) BrO events, such as are observed in the large satellite-detected BrO events associated with storms. The DF is a bulk average for the month, but there were very likely periods where the actual DF was larger, only to be reduced by return of bromide following deactiviation. A DF-based model would also not properly deal with mass transport limitations to aerosol particles that limit gas-surface reactions for supermicron aerosol particles. Thus, there would be a size dependence to aerosol reactivity that is not modeled by a simple DF-based approach. Lastly, it seems likely that there is a limit to how much reactive bromine can be produced, where the limit is likely related to the availability of radicals. This apparent limit is observed in the fact that few manuscripts report BrO mixing ratios more then ∼30-40 pptv. Presumably any such limit would affect the Antarctic DF data that are the basis for the seasonal DF model

being applied in this manuscript. If a DF-based scheme from the Antarctic were used and then the snow salinity (thus SSA mass concentration) is scaled by 3.5x, it would give 3.5x as much production of reactive bromine, but that may then exceed the ability to actually produce reactive bromine precursors from this SSA. The text also says that "... we could not justify this seasonal DF pattern, which demands further systematic measurements in the Arctic. As used in previous modeling studies, a non-seasonal (size dependent) DF scheme for the NH is used for comparison." I think the authors are saying that the process that they are using in this manuscript is not justifiable; if so, why did they describe the seasonal DF mechanism, and it seems they used it for most of the modeling.

The abstract (and text) read as if the "result" of attribution of bromine is a result of this study or that model simulations were modified to agree with some aspect of the observations, which doesn't appear to be the case. The model seems to turn on and off processes that are prescribed to get this attribution and no clear "inverse model­ing" was carried out to get this reported attribution. This wording should be clearer. In lines 27-29, the implication is made that blowing snow is the source of ODEs, but as described above, they have not uniquely shown that other sea-ice processes (e.g. snowpack chemistry, SSA production from open leads) cannot also be a source of ODEs. Therefore, this wording is misleading. I don't mean to differ with the idea that some ODEs may arise from blowing snow; I think they do, but the wording here dis­counts other literature-supported ideas for reactive halogen production by not testing them and also by believing that their parameterization is uniquely identified with they process they are trying to model. On lines 34-37, the abstract again tries to uniquely connect the improvement of agreement with observations with the process they in­tended to model (e.g. blowing snow). I expect that addition of other reactive halogen production models (e.g. snowpack and SSA from open leads) could also result in a model that "works reasonably well", so they have not shown that their improvement requires blowing snow.

Other issues:

The manuscript's writing is not precise and lacks significant details. The terms "bromine" is used very often, but in some cases they seem to be referring to reactive bromine (e.g. bromine radical species), total inorganic bromine (which is sometimes described as BrY, but often it is not clear), or sometimes seeming to include SSA bromide (Br-). The manuscript should chose a language for these species and use them consistently as opposed to the current confusing approach.

p2, line 15. VMR is not defined.

P2, line 26. The inclusion of "stratospheric BrO intrusions" is misleading here. Salawitch et al. 2010 discusses BrO total column enhancement due to stratospheric BrO intrusions, while this section is talking about release of reactive halogens in the lower troposphere. This should be clarified.

p2, line 34. "cruise data" – clarify. Note that this is Antarctic data.

p2, line 43. The text says "bromine depletion", when I think they mean "bromide depletion". This is one of many instances of inaccurate use of bromine-related terms. I am not pointing them all out, but they happen dozens of times throughout the text. Please chose terms, define them early and use them accurately.

p3, line 15. Specify "Pallas, Finland"

p3, line 24. What is an "integration"? Please explain.

p3, line 26. Is "validate" the right term here? Neither SSA production nor reactive bromine release is actually validated in this study – the effect of these modeled processes on ozone depletion and BrO are explored.

p4, line 7. The sentence "During the period..." is quite hard to read. I think "All" on line 8 maybe should not be capitalized?

p4, line 29. Cut "that" at the end of the line. Next line says "surface spectral reflectance

of 0.9" and is confusing. I think a non-spectral (e.g. not a function of wavelength) albedo of 0.9 is used in the spectral band of the retrieval (a band in the UV). If a spectrally varying albedo is used, please give more details describing this function.

p5, line 4. "...to the surface." is not well defined here. Earlier, the manuscript indicates that for VCD trop correction, BrO is assumed to be well mixed and below 400m. Thus this station (610m AGL) is above the BrO layer and looks up, so it has little sensitivity below it. Please clarify what is meant by "surface" in this context.

p5, line 15. 10 hPa is a pressure not an altitude, please be clearer.

p5, line 22. Has this model actually been validated in terms of vertical transport under stable Arctic conditions? I went to check these citations and found that "Ruti et al. 2011" is not in the citations, so I could not check. In another place, the text seems to indicate that the validation in Russo et al. 2011 is for tropical conditions. On page 6, it says "A detailed comparison of model characteristics in vertical mixing and transport of tropical boundary layer tracers was performed by Russo et al. (2011)". Please indicate if the model has been validated for the Arctic.

p5, line 26. Ozone deposition has been studied by Helmig, Bocquet, and others working in that group. Please include this work.

p6, line 38. How is "a full distribution of surface snow salinity" used in the model? Please clarify. Generally, details on how this model works are lacking.

p7, lines 1-5, also bottom of prior page. Why were these changes made? It appears that the model was being tuned in some way, but to what end? I believe that the Huang and Jaegle work has tuned the model in multiple ways so as to agree with various truth metrics (e.g. aerosol extinction from satellite, SSA observations, etc.). Please put your modifications into the context of other tunings and give a description of why this was done (e.g. what truth metric were you trying to match when you chose say N=20?).

p7, line 15. It is discussed that this approach "may introduce bias". Please describe

the model here, possibly pointing to a future "discussion" of this point.

p6, line 21. The language seems to often be in jargon of the field. What is a "dynamical core"?

p6, line 25. What does "but has since developed differently and now..." mean?

p6, line 28. I think it points to lack of detail on what the models are doing, but I don't understand what "tagged and tracked for online calculation of heterogenous reaction rates." means. Does the UKCA model not consider heterogenous reactions?

p6, line 36. Is there no heterogenous reaction of HOBr with aerosol bromide? If that process is considered, is the Br2 produced by it put in the grid cell (vertically) where it happened or is it placed at the surface?

p7, line 10. "model responses"? Maybe "model results"? This confusion continues onto line 11, where I think the species compared are bromine mixing ratio (or partial column) and ozone mixing ratio.

p7, line 12. maybe "model configuration" instead of "set-ups"?

p7, line 16. Many acronyms in this line. Can the point be made with words?

p7, line 27. I think the "minimum monthly-average VMRs" are being discussed. These are not the "minimum VMRs", which are clearly lower than the monthly average.

p7, line 32. I would say "further south".

p7, line 25. Again the use of imprecise language on bromine species is evident. I don't think they mean "Br" atoms on this line.

p7, line 37. I think they mean to say "OO-sourced bromine does not alter..."

p7, line 40. "maximum ozone loss" is confusing in this context. Maybe the peak of the annual ozone loss amount is meant?

p8, line 5. This finally defines BrY. Please move the definition much earlier and clarify

your bromine terms. I think the formula should say 2 times Br2. The X is confusing.

p8, line 29. 0.5 ppt of BrY is discussed here, which makes me wonder if BrY is the surface BrY, or includes BrY at higher altitudes? Please clarify what is being measured and how VSLS bromocarbons contribute to it.

p9, top. The manuscript does a good job describing the distributions of ozone observed and as calculated by the model, but the discussion of this point is lacking. The observed ozone distributions are sometimes bimodal, and are generally much flatter, while the modeled distributions are mono-modal and simply shifted to lower values. However, the general metric used for skill of the model is the monthly mean (e.g. Fig. 2). Even if the mean were correct, a differing distribution function indicates that the modeling is having problems reproducing the processes.

p9, line 27. The statement "In general, boundary layer ozone is influenced by column BrO...." I am not sure I understand if this is a result of the current study or an idea from the literature that is being cited (without reference) as a partial explanation of this study. Please clarify this section of the text.

p10, line 33. Please add "modeled" to make the text read "that affect modeled bromine emissions".

p10, line 37. Please consider Arctic snow data as discussed above.

p11, lines 1-4. This sentence is long and not clear. I'm not sure what is being discussed with respect to a "cut-off size". If there were actual heterogeneous processes being modeled, then the size distribution would matter a lot. Submicron aerosol particles tend to have little mass transport limitations for gas-surface reactions, but super-micron particles suffer transport limitations, so the same mass loading of SSA would have drastically different heterogeneous reactivity if it were all sub-micron or all super-micron. Please clarify and discuss.

p11, lines 10-14. In the presence of heterogenous reactions that release Br- from

aerosol particles and also formation of HBr that then sticks back to particles, the DF will be highly variable. Snowpack bromide observations (e.g. Krnavek et al., 2012) indicate that some snow is enhanced and some is depleted, and that varies in time. Thus, the use of a monthly DF seems unrealistic in the presence of gas-surface exchange of bromine species.

p11, lines 22-32. Again, other models of sea-ice-related production of reactive bromine (e.g. snowpack, SSA from open water leads) may be able to also explain the springtime ODEs and tropospheric BrO – this manuscript just did not test them. Thus, I agree that there is some sea-ice-related mechanism, but not necessarily only the blowing snow mechanism. The manuscript disregards other potential processes and thus may be misrepresenting the actual underlying physical process (or more likely multiple processes).

References:

Rachel M. Kirpes, Daniel Bonanno, Nathaniel W. May, Matthew Fraund, Anna J. Barget, Ryan C. Moffet, Andrew P. Ault, and Kerri A. Pratt, ACS Central Science 2019 5 (11), 1760-1767 DOI: 10.1021/acscentsci.9b00541

Krnavek, Laura & Simpson, William & Carlson, Daniel & Domine, Florent & Douglas, Thomas & Sturm, Matthew. (2012). The chemical composition of surface snow in the Arctic: Examining marine, terrestrial, and atmospheric influences. Atmospheric Environment. 50. 349–359. 10.1016/j.atmosenv.2011.11.033.

Pratt, K., Custard, K., Shepson, P. et al. Photochemical production of molecular bromine in Arctic surface snowpacks. Nature Geosci 6, 351–356 (2013). https://doi.org/10.1038/ngeo1779

Xu, W., Tenuta, M., and Wang, F. ( 2016), Bromide and chloride distribution across the snow‐sea ice‐ocean interface: A comparative study between an Arctic coastal marine site and an experimental sea ice mesocosm, J. Geophys. Res. Oceans, 121,

5535– 5548, doi:10.1002/2015JC011409.

---

## Author Comment (AC1) · 31 Jul 2020

**We thank reviewer #1 thoughtful comments on our manuscript. Below are our responses to each of the question raised.**

*Question: A major lack of the manuscript comes from the belief that the process trying to be model is uniquely represented by the model. This belief underlies statements such as in the abstract, where the manuscript states "...reproducing both ODEs and BEEs in the Arctic indicates that the relevant parameterizations implemented in the models work reasonably well, which supports the proposed mechanism of SSA and bromine production from blowing snow on sea ice." It is true that inclusion of this modeled process reduces springtime ozone and thus getting the model results closer to observations. However, to really make this statement (and many others like it in the manuscript), other models of sea-ice-related reactive bromine release would have need to have been considered and shown to be of lower skill than the proposed blowing snow SI- sourced bromine model. Without considering if other sea-ice-related processes could work effectively, the authors have not shown that this process is uniquely the one that is responsible. Other models in the literature could release of reactive bromine from snowpack (e.g. Pratt et al. 2013), production of SSA from wind over open water in sea ice leads between ice floes (e.g. Kirpes et al., 2019).*

Answer: In the past decades, more than half a dozen of different mechanisms were proposed as sources of reactive bromine in polar regions, including the snowpack and the open leads mechanisms as pointed by the reviewer. The aim of this study is to focus on one of them, which is the blowing-snow related SSA production scheme addressed in this manuscript and relevant published literatures. Investigating other processes or quantifying their relative contributions is not our goal and out of the research scope of this study. In this study, we demonstrated that blowing snow may be an important source of reactive bromine and models with this scheme implemented can reproduce well observed ozone and BrO. However, our finding does not necessary rule out other processes. To avoid misleading readers, in the revision, we clearly highlight this point in the Summary section: "The success of blowing snow mechanism does not necessarily rule out other possibilities, including the proposed candidates of reactive bromine from snowpack, open leads, frost flowers, sea ice surface, etc." With other words, we do not state that this process is uniquely the one that is responsible for ODE. We have been very careful in our wording on this subject. The present work shows that the some ODEs can be reproduced by models but also that not all events can be captured. We find that the result is a major step forward in our understanding of ODE.

*Question: To be devil's advocate, one could make a model of seasonally varying dry deposition of ozone to snowpack, which could be tuned to get agreement with the observed average monthly ozone data. Should the agreement of this model with average ozone observations then be taken as a sign that this process is the actual physical process that is occurring? The authors neither show that their model is unique nor do they present external validation of aspects of their model.*

Answer: Firstly, a certain level of tuning is unavoidable in modelling, however, adjusting a parameter must be based on either in situ measurement or laboratory data or be constrained by some other reasonable assumption. To follow the reviewer's analogy, if we "make" a model of seasonally varying dry deposition of ozone to snowpack and get the agreement of the model with average ozone observations by tuning dry deposition velocity, then we have shown that such a fictitious process is a candidate to explain our limited observations. Of course, it would then be eliminated by further observations, or by showing via the model that this implies unreasonable constrains on other parameters or processes. Note that in p-

TOMCAT, we decreased (not increased) the original ozone dry deposition velocity on snow/ice (in order to match the UKCA model's average velocity). Thus the simulated spring ozone depletion in the model is not due to the change of ozone dry deposition velocity. Secondly, monthly ozone data comparison is just one of the metrics used in this study to evaluate model's ability; the other one is time-series comparison of hourly ozone and daily tropospheric column BrO from GOME-2. It is clear that monthly data cannot tell the timing and duration of ODEs and BEEs observed in the spring, which hold the key to examine and effectively validate the processes implemented in the models. As shown in new Fig. 7-8 and old Table 3 (new Table 4), simulated 1-hour ozone in p-TOMCAT and the observations are significantly correlated with observed ozone, with $R$=0.68 at Tiksi, 0.49 at Barrow, 0.60 at Summit, 0.75 at VRS, and 0.22 at Alert. UKCA 3-hour ozone output shows a similar correlation for the same comparison with $R$ of 0.68 at Tiksi, 0.18 at Barrow, 0.47 at Summit, 0.62 at VRS and 0.41 at Alert (new Fig. 8, see inserted figure in our response to reviewer #2). Modelled $BrO_{trop}$ in the two models also shows significant correlation with the GOME-2 data in most cases (old Fig. 7 and 10, new Fig. 10-11 and 14-15). They strongly indicate that the physical and chemical processes implemented in the models work reasonably well. Apart from the evaluation in this study, we have compared modelled ozone and BrO with various observations in polar regions in many previous work (see Theys et al., 2009; Yang et al., 2010; Zhao et al., 2016, 2017; Legrand et al., 2017). In recent years, we have strictly evaluated the blowing snow related SSA production scheme (Levine et al., 2014; Rachael et al., 2016; 2017; Yang et al., 2019).

*Question: The use of Antarctic snow salinity data scaled by what appear to be a number of tuning parameters without Arctic validation does not give confidence in this SSA production model. Specifically, on page 5, line 37, the authors indicate that they have altered parameters of their model. Why? To what end? Why did they change the snow salinity by 3.5 times Antarctic data? The authors indicate on page 10, line 37 that "We lack snow data on Arctic sea ice to constrain the dataset used in this study". There are papers on snow on sea ice in the Arctic (e.g. Pratt et al. 2013, Krnavek et al., 2012, Xu et al., 2016). How do these observations compare with their scaled salinity.*

Answer:  In the revision, we added a new paragraph to discuss about this issue, see below:

"All parameters applied in this study for the Arctic SSA simulation are directly taken from our recent SSA modelling work by Yang et al. (2019), including a 3.5 times Antarctic snow salinity for the Arctic. The Antarctic Weddell Sea snow salinity is a surface snow salinity in distribution (Frey et al., 2019), which is different to the constant value (=0.3 psu, practical salinity unit) used in Legrand et al. (2016), Zhao et al. (2017) and Rhode et al. (2017). The trebled snow salinity assumption is taken from Yang et al. (2008) to reflect the likelihood that Arctic snow is more saline than in the Antarctic due to reduced precipitation. This assumption is partly justified by surface snow [Cl$^-$] concentrations observed at the two poles. For instance, an averaged surface snow (top 1-2 cm) [Cl$^-$] concentration of $368$ $\mu$M is derived from the Weddell Sea, Antarctic (https://ramadda.data.bas.ac.uk/repository/entry/show?entryid=853dd176-bc7a-48d4-a6be-33bcc0f17eeb, Frey et al. (2019)).  In the Arctic, Pratt et al. (2013) reported a mean surface [Cl$^-$] concentration of 1,121 $\mu$M (top 1 cm) over coastal sea ice near Barrow, Alaska, and Krnavek et al. (2012) reported a much higher surface [Cl$^-$] concentration of 21,058 $\mu$M over first-year sea ice and 63,217 $\mu$M over multi-year sea ice over a slightly deeper depth of 2~3 cm below the surface. They are about 3, 57 and 172 times of the Weddell Sea surface salinity." The relative higher salinity in the Arctic is partly related to less precipitation as

already mentioned. For instance, the depth of snowpack on sea ice near Barrow, Alaska is in a range of 10-40 cm (Krnavek et al., 2012), while in the Weddell Sea, the mean snow depth over FYI is 20.9 cm and 50.0 cm over MYI (Frey et al., 2019).

To investigate the effect of snow salinity uncertainty on ozone and Br$_Y$, we performed two model experiments: pTOMCAT_high_salinity applying a 10 times Weddell Sea salinity and pTOMCAT_low_salinity applying a 1 times Weddell Sea salinity (new Table 3 shown below. Note that there are total 8 additional experiments listed in the table). Results are shown in new Fig. 16 (shown below) and new Table 3. Comparing to pTOMCAT_control run, derived sea-ice sourced BrY (April) in pTOMCAT_high_salinity run increases by +94.8%, corresponding to additional ozone loss by -37.5%. Similarly, sea-ice sourced BrY in pTOMCAT_low_salinity decreases by -60%, corresponding to ozone gain by +61%. See further discussion in section 5: Discussions (see our reply to reviewer #2).

New Table 3: Model sensitive experiments. The parameters involved in the experiments are listed in the 2$^{nd}$ column. The derived sea-ice-sourced Br$_Y$ and ozone change (relative to pTOMCAT_OO_VSLS run) is in the 3$^{rd}$ and 5$^{th}$ column, respectively. The corresponding percentage (and change) of BrY and ozone are in the 4$^{th}$ and 6$^{th}$ column, respectively. The ozone difference (also relative to pTOMCAT_OO_VSLS run) is in the 5$^{th}$ column, with percentage of the control run result (and change) in the 6$^{th}$ column. The values are for April, 2007 and representing average of all the six sites.

| Experiments | Key parameters | $\Delta Br_Y$ (pptv) in April (relative to pTOMCAT_OO_VSLS) | % of pTOMCAT_control $\Delta Br_Y$ (and difference) | $\Delta O_3$ (ppbv) in April (relative to pTOMCAT_OO_VSLS) | % of pTOMCAT_control $\Delta O_3$ (and difference) |
|---|---|---|---|---|---|
| pTOMCAT_control | N=20 shape parameter α=3 scale parameter β=37.5 μm dm$_i$/dt=$d_i$ snow salinity =3.5× Weddell Sea value cut-off radius (dry NaCl)=10 μm | 47.2 | 100 (0) | -20.4 | 100 (0) |
| pTOMCAT_spectrum_1 | same as pTOMCAT_control but N=10 | 37.8 | 79.9 (-20.1) | -17.0 | 83.5 (+16.5) |
| pTOMCAT_spectrum_2 | same as pTOMCAT_control but N=1, α=2, β=70 μm and dm$_i$/dt=constant | 27.3 | 57.8 (-42.2) | -12.5 | 61.1 (+38.6) |
| pTOMCAT_low_salinity | same as pTOMCAT_control but snow salinity =1× Weddell Sea value | 18.9 | 40.0 (-60.0) | -8.0 | 39.0 (+61.0) |
| pTOMCAT_high_salinity | same as pTOMCAT_control but snow salinity =10× Weddell Sea value | 92.0 | 194.8 (+94.8) | -28.1 | 137.5 (-37.5) |
| pTOMCAT_2×DF | same as pTOMCAT_control but 2×DF* | 87.1 | 184.4 (+84.4) | -27.7 | 135.7 (-35.7) |
| pTOMCAT_0.5×DF | same as pTOMCAT_control but 0.5×DF | 24.9 | 52.7 (-47.3) | -11.1 | 54.2 (+45.8) |
| pTOMCAT_SSA20μm | same as pTOMCAT_control run but cut-off radius=20 μm | 67.3 | 142.3 (+42.3) | -24.7 | 121.1 (-21.1) |
| pTOMCAT_SSA5μm | same as pTOMCAT_control but cut-off radius=5 μm | 21.2 | 44.9 (-55.1) | -9.1 | 44.6 (+45.4) |

*: if the 2*DF value is > 1.0, then a maximum value of 1.0 is used.

[Figure]

New Fig. 16: Model sensitivity experiments results. Note that ozone observations are climatology and model outputs are only for year 2007.

*Question: It is not clear from this study how reactive bromine is produced from the SSA bromide. The section on this topic is 3.1, and also refers to the supplemental table 1. First, it is not clear what the model is doing. Is the model actually considering heterogeneous chemical reactions that would convert SSA bromide (Br-) to gas-phase reactive bromine precursors (e.g. Br2)? I expect that the model is not actually considering these reactions, but instead simply multiplying the DF * SSA to get "lost" Br- from the SSA and making this into Br2. The manuscript should be clear as to what is actually being modeled. If the process is simply taking the bulk DF times SSA bromide, then there are a number of physical and chemical problems with this approach. For example, we know that reactive bromine deactiviates fairly rapidly (e.g. reaction of Br with H2CO to form HBr), which then partitions to the aerosol particle phase, increasing Br- in the SSA (and thus reducing the DF). This recycling of reactive bromine is needed for persistent (e.g. multi-day) BrO events, such as are observed in the large satellite-detected BrO events associated with storms. The DF is a bulk average for the month, but there were very likely periods where the actual DF was larger, only to be reduced by return of bromide following deactiviation. A DF-based model would also not properly deal with mass transport limitations to aerosol particles that limit gas-surface reactions for supermicron aerosol particles. Thus, there would be a size dependence to aerosol reactivity that is not modeled by a simple DF-based approach.*

Answer: In section 2.1.2.3 "Treatment of Bromine Emission From Sea Salt" of Yang et al. (2005), we have justified the use of DF to describe reactive bromine release from SSA. The

use of bulk depletion factor measured to constrain the total amount of bromide that can be released from SSA has been proven to be a convenient method in dealing with SSA-sourced reactive bromine (see Yang et al., 2005; 2010; Breider et al., 2012; Parrella et al., 2012), even though the actual DF for individual salt particles is different and varying with time and location. Heterogeneous recycling of gas-phase bromine species is included, including the HBr+HOBr→ Br$_2$ reactivations, see details in section 2.5 "Treatment of reactivation of inactive bromine species" of Yang et al. (2010). Note that, in our modelling approach, we manually "separate" the above two processes by dealing with net bromide release and the following heterogeneous recycling independently. Thus, the recycling reaction will not allow particles to net gain or lose bromine (see detail on how to deal with this issue in Yang et al., 2010). A size-dependent DF scheme (supplementary Table1) is applied in our modelling for comparison, see new (and old) Fig. S6. Finally, in order to test our model sensitivity to the DF parameter, in the revision, we performed two experiments by doubling and halving the DF value used in the control run (see new Table 3 above). Comparing to the pTOMCAT_control run, the sea-ice sourced Br$_Y$ (April) in pTOMCAT_2×DF run increases by +84.4%, corresponding to additional ozone loss by -35.7%. The sea-ice sourced BrY in pTOMCAT_0.5XDF reduces by -47.3%, corresponding to ozone increase by +45.8%. Given that the observed DF values in March, April and May are close to 0.5 (Table 1), the doubled DF means a unit DF=1 is used, indicating that all bromide in SSA is emitted to the air as a source of reactive bromine. Thus, this is an extreme experiment representing the maximum effect from blowing snow sourced SSA.

*Question: Lastly, it seems likely that there is a limit to how much reactive bromine can be produced, where the limit is likely related to the availability of radicals. This apparent limit is observed in the fact that few manuscripts report BrO mixing ratios more then 30-40 pptv. Presumably any such limit would affect the Antarctic DF data that are the basis for the seasonal DF model being applied in this manuscript. If a DF-based scheme from the Antarctic were used and then the snow salinity (thus SSA mass concentration) is scaled by 3.5x, it would give 3.5x as much production of reactive bromine, but that may then exceed the ability to actually produce reactive bromine precursors from this SSA.*

Answer: As demonstrated previously, surface snow salinity in the Arctic is likely higher than that in the Antarctic. The logistic issue relevant is that under a high snow salinity, the corresponding dry NaCl size formed will be larger than that under a low snow salinity. For a 3.5 times salinity, the dry NaCl radius formed will be larger by ~50%. Since not all SSA formed can release bromide, the application of a cut-off threshold size (a dry NaCl radius of 10 μm in the control run) means SSA in size larger than that threshold size will not be allowed to act as a source of reactive bromine. Thus, bromide release from blowing snow is not linear to snow salinity change. Our model output shows that for a ~3 times increase in snow salinity (pTOMCAT_high_salinity vs pTOMCAT_control), sea-ice sourced Br$_Y$ (April) in the Arctic only increases by a factor of ~1 (94.8% in new Table 3). The response of reactive bromine released from SSA varies and depends on the snow salinity. The potentially high snow salinity in the Arctic indeed implies that there will be relatively more SSA formed in the Arctic than in the Antarctic from same amount of sublimated water in blowing snow, and may induce more reactive bromine release.

*Question: The text also says that "... we could not justify this seasonal DF pattern, which demands further systematic measurements in the Arctic. As used in previous modeling studies, a non-seasonal (size dependent) DF scheme for the NH is used for comparison." I think the authors are saying that the process that they are using in this manuscript is not*

*justifiable; if so, why did they describe the seasonal DF mechanism, and it seems they used it for most of the modeling.*

Answer: Our use of the word "justify" is unfortunate; we meant "validate". Of course, we did meant to say that the DF processes that is not justifiable, just that we lack data to establish its validity. It is a real shame there is no seasonal DF available in the Arctic after so many years of in situ measurements. In the absence of data to confirm (or refute) our model, why derive a seasonal DF pattern for the Arctic based on the available seasonal DF from Antarctica. It is the best we can do; we are simply acknowledging that cannot justify the derived seasonal DF. In addition, in this study, we also perform a comparison to the model run with a fixed DF for all seasons. The major aim of applying a seasonal DF to the Arctic is to investigate how to improve the model-data agreement in Arctic surface ozone reproduction. As we clearly mentioned in the manuscript, this work demonstrates further systematic measurement of this key factor in the Arctic is necessary.

*Question: The abstract (and text) read as if the "result" of attribution of bromine is a result of this study or that model simulations were modified to agree with some aspect of the observations, which doesn't appear to be the case. The model seems to turn on and off processes that are prescribed to get this attribution and no clear "inverse modeling" was carried out to get this reported attribution. This wording should be clearer. In lines 27-29, the implication is made that blowing snow is the source of ODEs, but as described above, they have not uniquely shown that other sea-ice processes (e.g. snowpack chemistry, SSA production from open leads) cannot also be a source of ODEs. Therefore, this wording is misleading. I don't mean to differ with the idea that some ODEs may arise from blowing snow; I think they do, but the wording here dis- counts other literature-supported ideas for reactive halogen production by not testing them and also by believing that their parameterization is uniquely identified with they process they are trying to model. On lines 34-37, the abstract again tries to uniquely connect the improvement of agreement with observations with the process they in- tended to model (e.g. blowing snow). I expect that addition of other reactive halogen production models (e.g. snowpack and SSA from open leads) could also result in a model that "works reasonably well", so they have not shown that their improvement requires blowing snow.*

Answer: We derive our conclusion that the blowing snow mechanism is sufficient to reproduce Arctic ODEs, but this is relative to the other two tropospheric sources of bromine (VSLS and open ocean sea spray) investigated in this study. Testing other mechanisms and quantifying their relative importance are beyond the scope of this study. In the revision, we added the following sentence in the summary: "The success of blowing snow mechanism does not necessarily rule out other possibilities, including the proposed candidates of reactive bromine from snowpack, open leads, frost flowers and sea ice surface, etc." In the abstract, we also added one sentence: "Note that this work dose not necessary rule out other possibilities that may act as a source of reactive bromine from sea ice zone."

*Question: The manuscript's writing is not precise and lacks significant details. The terms "bromine" is used very often, but in some cases they seem to be referring to reactive bromine (e.g. bromine radical species), total inorganic bromine (which is sometimes described as BrY, but often it is not clear), or sometimes seeming to include SSA bromide (Br-). The manuscript should chose a language for these species and use them consistently as opposed to the current confusing approach.*

Answer: Points accepted. In the revision, we used more precise terms to avoid confusion.

Question: p2, line 15. VMR is not defined.

Answer: Definition of VMR is given in its first place.

*Question: P2, line 26. The inclusion of "stratospheric BrO intrusions" is misleading here. Salawitch et al. 2010 discusses BrO total column enhancement due to stratospheric BrO intrusions, while this section is talking about release of reactive halogens in the lower troposphere. This should be clarified.*

Answer: In the revision, we clearly mentioned that "In addition, stratospheric BrO intrusions in association with downward transport of air masses from lower stratosphere could affect polar free tropospheric BrO (Salawitch et al., 2010)."

*Question: p2, line 34. "cruise data" – clarify. Note that this is Antarctic data.*

Answer: We added ", Antarctic" after "the Weddell Sea"

*Question: p2, line 43. The text says "bromine depletion", when I think they mean "bromide depletion". This is one of many instances of inaccurate use of bromine-related terms. I am not pointing them all out, but they happen dozens of times throughout the text. Please chose terms, define them early and use them accurately.*

Answer: We have gone through the whole text and corrected the misusing of the terms.

*Question: p3, line 15. Specify "Pallas, Finland"*

Answer: Done.

*Question: p3, line 24. What is an "integration"? Please explain.*

Answer: "integration" means "simulation", a term widely used in modelling related studies.

*Question: p3, line 26. Is "validate" the right term here? Neither SSA production nor reactive bromine release is actually validated in this study – the effect of these modeled processes on ozone depletion and BrO are explored.*

Answer: We re-phrased this sentence: "to validate the effect of these modelled processes on ozone depletion and BrO enhancement".

*Question: p4, line 7. The sentence "During the period..." is quite hard to read. I think "All" on line 8 maybe should not be capitalized?*

Answer: In the revision, we rewrote this sentence, it reads now: "During the period of interest here, all ozonesondes used were electrochemical concentration cells (ECC) (Komhyr, 1969), manufactured by Environmental Science (EN-SCI) Corp. All sondes used the conventional neutral-buffered 1% potassium iodide sensing solution."

*Question: p4, line 29. Cut "that" at the end of the line. Next line says "surface spectral reflectance of 0.9" and is confusing. I think a non-spectral (e.g. not a function of wavelength) albedo of 0.9 is used in the spectral band of the retrieval (a band in the UV). If a spectrally varying albedo is used, please give more details describing this function.*

Answer: Point taken. In the revision, we removed the misleading word "spectral" in the sentence.

*Question: p5, line 4. "...to the surface." is not well defined here. Earlier, the manuscript indicates that for VCD trop correction, BrO is assumed to be well mixed and below 400m. Thus this station (610m AGL) is above the BrO layer and looks up, so it has little sensitivity below it. Please clarify what is meant by "surface" in this context.*

Answer: In the revision, we have the new sentence "The retrievals were performed on a 0-4 km altitude grid with 0.2 km resolution. Due to the altitude of the instrument (610 m) and the lack of low or negative elevation angles, the retrieved profiles are only sensitive to well-mixed BrO in a deep boundary layer, and to lofted BrO event."

*Question: p5, line 15. 10 hPa is a pressure not an altitude, please be clearer.*

Answer: We added "(~31km)" after "10 hpa" for an easy understanding.

*Question: p5, line 22. Has this model actually been validated in terms of vertical transport under stable Arctic conditions? I went to check these citations and found that "Ruti et al. 2011" is not in the citations, so I could not check. In another place, the text seems to indicate that the validation in Russo et al. 2011 is for tropical conditions. On page 6, it says "A detailed comparison of model characteristics in vertical mixing and transport of tropical boundary layer tracers was performed by Russo et al. (2011)". Please indicate if the model has been validated for the Arctic.*

Answer: Sorry, "Ruti et al. 2011" should be "Hoyle et al, 2011". They are both for tropical conditions. There is no such kind of modelling validation in the Arctic.

*Question: p5, line 26. Ozone deposition has been studied by Helmig, Bocquet, and others working in that group. Please include this work.*

Answer: We cited the review article by Helmig et al. (2007).

*Question: p6, line 38. How is "a full distribution of surface snow salinity" used in the model? Please clarify. Generally, details on how this model works are lacking.*

Answer: To avoid confusion, we changed 'a full distribution" to "a probability of surface snow salinity" as used in Fig. 16b of Frey et al. (2019).

*Question: p7, lines 1-5, also bottom of prior page. Why were these changes made? It appears that the model was being tuned in some way, but to what end? I believe that the Huang and Jaegle work has tuned the model in multiple ways so as to agree with various truth metrics (e.g. aerosol extinction from satellite, SSA observations, etc.). Please put your modifications into the context of other tunings and give a description of why this was done (e.g. what truth metric were you trying to match when you chose say N=20?).*

Answer: As clearly mentioned previously that all the parameters applied in this study for SSA production are directly taken from recent SSA modelling work by Yang et al. (2019); thus there is no further tuning in this study. The different parameters mentioned in the manuscript are for track record only. The reason why we chose this set of parameter (namely the SI_Classic_BX20 run in Yang et al. (2019) is that it is one of the parameter sets that best matched the Weddell Sea SSA in size range of 0.4 – 10 μm. This set also gave the highest SSA mass loading in Polar Regions (Yang et al., 2019). We have clearly mentioned this issue in the revision. To investigate model sensitivity to the SSA spectrum, we performed two additional model experiments: pTOMCAT_stectrum_1 and pTOMCAT_stectrum_2 (new Table 3). The former one applies same parameter set but a small N=10, the later one corresponds to the SI_Base run in Yang et al., (2019), another "best" one marches the cruise data. As shown in new Table 3 that the sea-ice sourced $Br_Y$ in these two runs reduces by -20.1% and ~42.2%, respectively, corresponding to ozone increase by 16.5% and 38.6%.

*Question: p7 line 15. It is discussed that this approach "may introduce bias". Please describe the model here, possibly pointing to a future "discussion" of this point.*

Answer: (I think you meant p6 line 15). In the revision, we added one sentence "Thus, we may overestimate the open-ocean-sourced SSA effect in polar regions, as the alkaline buffering effect is not considered."

*Question: p6, line 21. The language seems to often be in jargon of the field. What is a "dynamical core"?*

Answer: "dynamical core" refers to the meteorology field used in the UKCA model and is generated by the "core" of the Unified Model. UKCA is just a chemistry-aerosol mode of the UK Earth System Model.

*Question: p6, line 25. What does "but has since developed differently and now…" mean?*

Answer: To avoid confusion, we simply deleted "but has since developed differently and now".

*Question: p6, line 28. I think it points to lack of detail on what the models are doing, but I don't understand what "tagged and tracked for online calculation of heterogenous reaction rates." Means. Does the UKCA model not consider heterogenous reactions?*

Answer: UKCA does contain heterogeneous reactions. UKCA used monthly climatology aerosol data for the heterogeneous rate calculation, and p-TOMCAT used online calculated SSA and cloud droplets for the heterogeneous reaction rate calculation. There are 21 size bins for SSA in generated from open ocean and sea ice, thus we have to "tag" them in order to track and record their concentrations in all grid cell at every time step.

*Question: p6, line 36. Is there no heterogenous reaction of HOBr with aerosol bromide? If that process is considered, is the Br2 produced by it put in the grid cell (vertically) where it happened or is it placed at the surface?*

Answer: Both models contain same heterogeneous reaction scheme as described in Yang et al. (2010), including the HOBr+HBr -->$Br_2$. This reaction happens in all grid cells where there are SSA and/or particles. Since we separate bromine emission from the heterogeneous

reaction on aerosols, thus emitted $Br_2$ from freshly generated SSA is only placed in the surface layer. Namely, once we used DF to describe the reactive bromine release from SSA, theses SSA will no longer allow net bromide release (or net gain of bromine) and only serve as a medium for heterogeneous reaction (see our explanation above).

*Question: p7, line 10. "model responses"? Maybe "model results"? This confusion continues onto line 11, where I think the species compared are bromine mixing ratio (or partial column) and ozone mixing ratio.*

Answer: OK, points taken.

*Question: p7, line 12. Maybe "model configuration" instead of "set-ups"?*

Answer: Point taken.

*Question: p7, line 16. Many acronyms in this line. Can the point be made with words?*

Answer: We deleted the sentence in response to reviewer #2's comment.

*Question: p7, line 27. I think the "minimum monthly-average VMRs" are being discussed. These are not the "minimum VMRs", which are clearly lower than the monthly average.*

Answer: We removed "minimum"

*Question: p7, line 32. I would say "further south".*

Answer: Point accepted.

*Question: p7, line 25. Again the use of imprecise language on bromine species is evident. I don't think they mean "Br" atoms on this line.*

Answer: *(actually it is p7 line 35)* We changed 'Br' to 'reactive bromine'

*Question: p7, line 37. I think they mean to say "OO-sourced bromine does not alter…"*

Answer: point accepted.

*Question: p7, line 40. "maximum ozone loss" is confusing in this context. Maybe the peak of the annual ozone loss amount is meant?*

Answer: Point accepted.

*Question: p8, line 5. This finally defines BrY. Please move the definition much earlier and clarify your bromine terms. I think the formula should say 2 times Br2. The X is confusing.*

Answer: Point accepted. We moved forward the definition of BrY.

*Question: p8, line 29. 0.5 ppt of BrY is discussed here, which makes me wonder if BrY is the surface BrY, or includes BrY at higher altitudes? Please clarify what is being measured and how VSLS bromocarbons contribute to it.*

Answer: As shown in new Fig.S7 (below), VSLS contribution to Arctic tropospheric Br$_Y$ increases from near surface layer ~0.5 pptv (in April and July) to ~2 pptv at ~200 hpa. In April, it only accounts for 2~4% of the total Br$_Y$ in the surface layer and ~40% at 200 hpa. In July, it accounts for 15~20% Br$_Y$ in the surface layer and >60% at 300-400 hpa.

[Figure]

New figure S7: Zonal mean of VSLS Br$_Y$ (pptv) and percentage contribution to total Br$_Y$ in April (a, b) and July (c, d). Values are from a 3-years p-TOMCAT integration.

*Question: p9, top. The manuscript does a good job describing the distributions of ozone observed and as calculated by the model, but the discussion of this point is lacking. The observed ozone distributions are sometimes bimodal, and are generally much flatter, while the modeled distributions are mono-modal and simply shifted to lower values. However, the general metric used for skill of the model is the monthly mean (e.g. Fig. 2). Even if the mean were correct, a differing distribution function indicates that the modeling is having problems reproducing the processes.*

Answer: The reviewer's statement that "the general metric used for skill of the model is the monthly mean" is not correct. Monthly mean is just one of the comparison metrics used in this study. Another important metric for skill of the model is the time series comparison for hourly surface ozone and daily tropospheric column BrO, in particular the statistical analysis between model output and observations as shown in new Fig. 7-10 and 14-15 (also see our answer to your similar question in page 2).

*Question: p9, line 27. The statement "In general, boundary layer ozone is influenced by column BrO...." I am not sure I understand if this is a result of the current study or an idea from the literature that is being cited (without reference) as a partial explanation of this study. Please clarify this section of the text.*

Answer: This conclusion is a result of this study, and mainly taken from the model output shown in new Table 4 (old 3). For example, observed ozone at Barrow and Alert are significantly correlated with modelled tropospheric column BrO but surface BrO. This is because ozone has much longer lifetime than BrO, due to vertical mixing and/or air ventilation at the top of the boundary layer, ozone and reactive bromine in the free troposphere may influence surface ozone within the boundary layer.

*Question: p10, line 33. Please add "modeled" to make the text read "that affect modeled bromine emissions".*

Answer: We have re-written this section.

*Question: p10, line 37. Please consider Arctic snow data as discussed above.*

Answer:  See our previous response to your similar question raised before.

*Question: p11, lines 1-4. This sentence is long and not clear. I'm not sure what is being discussed with respect to a "cut-off size". If there were actual heterogeneous processes being modeled, then the size distribution would matter a lot. Submicron aerosol particles tend to have little mass transport limitations for gas-surface reactions, but super- micron particles suffer transport limitations, so the same mass loading of SSA would have drastically different heterogeneous reactivity if it were all sub-micron or all super- micron. Please clarify and discuss.*

Answer: The 'cut-off size' was introduced to reflect the observations (see Sander et al., 2003) that large giant SSAs have relatively lower bromide depletion strength (due to relatively shorter lifetime) than small SSAs. In the control run, we used a dry NaCl radius of 10 μm as the threshold size, which means a large SSA particle with dry NaCl radius larger than 10 μm will not be considered as a source of reactive bromide. On the other hand, the choice of the cut-off threshold size also affects the emission flux of reactive bromine from SSA. In the revision, we performed two extra experiments: pTOMCAT_SSA20, in which a large threshold radius of 20 μm is used, and pTOMCAT_SSA5, in which a small threshold radius of 5 μm is used. See results shown in in new Fig. 16 and new Table 3, more in the discussion section.

*Question: p11, lines 10-14. In the presence of heterogenous reactions that release Br- from aerosol particles and also formation of HBr that then sticks back to particles, the DF will be highly variable. Snowpack bromide observations (e.g. Krnavek et al., 2012) indicate that some snow is enhanced and some is depleted, and that varies in time. Thus, the use of a monthly DF seems unrealistic in the presence of gas-surface exchange of bromine species.*

Answer: As mentioned previously that we applied a non-process based approach to deal with the bromide release from SSA, namely a bulk rather than a single-particle based approach. Thus we do not resolve every single SSA but focus on their average (Yang et al., 2005).  In order to investigate the uncertainty this method induces, we performed two model experiments: pTOMCAT_2XDF used a doubled DF, and pTOMCAT_0.5XDF used a halved DF (new Table 3). Given that the original DF values in March, April and May are close to 0.5 (Table 1), the doubling scenario means DF is close to 1, representing an extreme case that all bromide in sea-ice sourced SSA will release to the air to act as a source of reactive bromine. Results are shown in new Fig. 16 and Table 3, and discussed in the discussion section.

*Question: p11, lines 22-32. Again, other models of sea-ice-related production of reactive bromine (e.g. snowpack, SSA from open water leads) may be able to also explain the springtime ODEs and tropospheric BrO – this manuscript just did not test them. Thus, I agree that there is some sea-ice-related mechanism, but not necessarily only the blowing snow mechanism. The manuscript disregards other potential processes and thus may be misrepresenting the actual underlying physical process (or more likely multiple pro- cesses).*

Answer: Point taken. In the revision, we clearly mentioned that this study does not rule out other possibilities.

---

## Author Comment (AC2) · 31 Jul 2020

**We thank reviewer #2 thoughtful comments on our manuscript. Below are our responses to each of the question raised.**

*Question: A weak aspect is the exploitation of the two different models. It seems a lot of effort to run both models for the analysis, yet not very much is made of comparing their performance and discussing possible reasons for their different performance or what can be learned from this. In many sections and figures, only one of the two models is shown (different single models for different sections), which is not well justified.*

Answer: In the revision, we showed both model results for all the cases discussed in the manuscript. Therefore, we added some new plots which are new Fig. 8, 10, 13, 15 (shown below).

*Question: Page 3, paragraph 2: There have in fact been multi-model assessments of Arctic sur- face ozone in global CTMs. See Monks et al., (2015), Emmons et al., (2015), and an older study by Shindell et al., (2008). Monks et al and Shindell et al both show over-prediction of surface ozone at Barrow in spring, likely as a result of missing halogen chemistry. However, Emmons et al show a general model \*underprediction\* over the depth of the Arctic troposphere in April compared with ozone sondes, suggesting that the halogen-induced bias may not be pervasive in the Arctic troposphere. It would be helpful to see these previous studies highlighted in the text for context.*

Answer: In the revision, we added the following sentences:

"Previous multi-model assessments of Arctic surface ozone in global chemistry transport models (CTMs) gave quite different implications on the role of halogens. For instance, Monks et al. (2015) and early modelling work by Shindell et al. (2008) both showed over-prediction of surface ozone at Barrow in spring, implying a result of missing halogen chemistry. However, Emmons et al. (2015) showed a general model under-prediction in April compared with ozone sondes, suggesting that the halogen-induced bias may not be pervasive in the Arctic troposphere. Thus, relatively little is known about model skill in reproducing polar spring boundary layer ozone, on time scales of hourly, daily and monthly, leaving a large gap in our understanding of the global ozone budget in the polar regions."

*Question: Page 5, line 3: "The retrievals were performed on a 0-4 km grid with 0.2 km resolution." Not clear what this means. What is a 0-4km grid?*

Answer: In the revision, we have the new sentence "The retrievals were performed on a 0-4 km altitude grid with 0.2 km resolution. Due to the altitude of the instrument (610 m) and the lack of low or negative elevation angles, the retrieved profiles are only sensitive to well-mixed BrO in a deep boundary layer, and to lofted BrO event."

*Question: Page 5, line7: Both models are driven by ERA-Interim data. For UKCA, please briefly explain what this means for the climate model. i.e. does this imply nudging with a certain degree of relaxation? Over what altitude range? It is important to recognise that this is different from a purely offline model (such as pTOMCAT). What else is prescribed / free-running between the models? Clouds? Surface fluxes?*

Answer: In the revision, we added the following two paragraphs to highlight the model differences in atmospheric dynamics:

"A global chemistry transport model, p-TOMCAT, and a global chemistry climate model, UKCA, are used in this study. The offline p-TOMCAT used 6-hour ERA-Interim dataset to drive its winds, temperature and moisture. The ERA-Interim data were taken from the European Centre for Medium-Range Weather Forecasts (ECMWF) (Dee et al., 2011). In this study, a nudged UKCA version is used to ensure a model meteorological field close to the real situation for data-model comparison. We follow the work of Telford et al. (2008) with a standard nudging relaxation parameter $G=1/6$ h$^{-1}$, which value lies within the range of relaxation parameters used by other models (Jeuken et al., 1996; Hauglustaine et al., 2004; Schmidt et al., 2006). We used the 6-hour ERA-Interim winds and temperature to constrain UKCA model's dynamical field. However, nudging is not applied to all levels; no nudging being applied above level 50 (~48 km), or below level 12 (~2.9 km (the actual height varies depending on the orography). To avoid instability of the model, moisture is not nudged to reanalysis data, therefore it is free running.

Both models applied a non-local boundary layer mixing scheme, but p-TOMCAT based on the parameterisation of Holtslag and Boville (1993), while UKCA based on the scheme of Lock et al. (2000). In terms of convective mass flux, p-TOMCAT applied the scheme of Tiedtke (1989) – which has been updated to increase convective transport to the mid and upper troposphere (Barret et al., 2010; Feng et al., 2011), and UKCA applied the bulk convection model of Gregory and Rowntree (1990). As shown in a multi-model inter-comparison in tropics, these two models showed different behaviour in terms of deep convective transport of tropical boundary layer tracers (Hoyle et al., 2011). The clouds and precipitation schemes are also different between the two models (Russo et al., 2011), resulting in different wash-out rates for aerosols and soluble chemical compounds. The precipitation bias in the op-TOMCAT model (Giannakopoulos et al., 2004) is remedied by applying a correction to force the simulated precipitation values towards Global Precipitation Climatology Project (GPCP) observations (Adler et al., 2003), following the work in Legrand et al. (2016). This corrected precipitation scheme has been used in recent sea salt aerosol modelling works (Rhodes et al., 2017; Yang et al., 2019). However, precipitation in UKCA is free running, therefore the two models may have different wet removal rates for soluble gaseous-phase species. Details of other model configurations, mainly in chemistry scheme used are described in sections 3.1 for p-TOMCAT and 3.2 for UKCA."

*Question: Page 5, line 23: Is the Law et al., (2000) study the most up-to-date reference for the model chemistry scheme? How up-to-date is the kinetic data used? How do these data compare with that used in the UKCA model for the same tropospheric reactions? Does p-TOMCAT include non-halogen related heterogeneous chemistry (it seems that UKCA does)-e.g. N2O5 hydrolysis on aerosol, which is likely important for winter / early spring ozone and NOy in the Arctic. Given the focus on comparing ozone performance between the models, it is important to acknowledge any important differences in the chemical schemes of the models.*

Answer: We added the following two paragraphs Section 3.1 to highlight he chemistry differences between the two model.

"The ozone photochemistry scheme applied to the model has been detailed in previous studies (Law et al., 1998, 2000) and Savage et al. (2004), with updates including an isoprene chemistry scheme, same as the one implemented to the UKCA model by Young et al. (2009) according to the method of Poschl et al. (2000), a hydrolysis reaction of $N_2O_5$ on aerosols and cloud droplets (Yang et al., 2005), a tropospheric bromine scheme involving both gaseous-phase reactions (Yang et al., 2005) and heterogeneous reactions (Yang et al., 2010), and a

Fast-J photolysis scheme developed by Voulgarakis et al. (2009b), which is not used in this study. They found that $N_2O_5$ hydrolysis can cause net $NO_X$ loss at high latitudes by up to 60% in the northern hemisphere and ~80% in the southern hemisphere (Yang et al., 2005). They found that including halogen-related heterogeneous reactions on aerosols and cloud droplet can significantly increase polar BrO partitioning by a factor of ~3 (Yang et al., 2010). This heterogeneous reaction scheme for halogen reactivation was also implemented to the UKCA model (Yang et al., 2014; Dennison et al., 2019; Ming et al., 2020)."

Ozone is dry-deposited in the bottom model layer with dry deposition velocity inferred from the study of Ganzeveld and Lelieveld (1995) by Giannakopoulos (1998). The original dry deposition velocity over ocean and snow (=0.05 cm s$^{-1}$) is reduced to 0.01 cm s$^{-1}$ in this study following recent modelling work by Hardacre et al. (2015) and Luhar et al. (2018) as well as the Helmig et al. (2007). Since p-TOMCAT only covers part of the stratosphere with a top layer height of ~31 km, a simplified stratospheric chemical scheme has to be used, including a pre-prescribed top boundary condition for ozone. Therefore, p-TOMCAT model is quite different from the UKCA model in upper troposphere and lower stratosphere. However, it is unlikely that the downwards transport of air mass in the polar region may significant influence near surface bromine."

And in section 3.2, we added one sentence: "In p-TOMCAT model, heterogeneous reactions occur also on cloud droplets, but UKCA does not include such reactions. Therefore, in free troposphere, the BrO partitioning in UKCA may be lower than that in p-TOMCAT, which may result in more soluble inorganic bromine species being washed-out by precipitation in UKCA, as discussed in section 4." In section 4 we added one sentence: "In addition, p-TOMCAT considers heterogeneous reactions on cloud droplets while UKCA does not, this difference may explain why BrO partitioning in p-TOMCAT is higher than that in UKCA, especially in free troposphere, where BrO partitioning can be as large as 50% (Fig. S2). In addition, the higher BrO partitioning in p-TOMCAT also attribute less BrY removal by dry and wet depositions."

*Question: Page 7, line 15: Care needs to be taken in over-interpreting the reason for differences between the models and assigning this to mainly physical parameters (and I agree that such differences between a climate model and CTM would be expected - although see also my comment on nudging), as it may be that there are important differences between the model chemical schemes (see point above). This is why it would be useful to point out more about these potential differences. I am not sure it is the case that only or a dominance of physical factors can be assumed.*

Answer: Point taken. We remove the relevant sentence (below) in the revision: "Given that both models used in this study share a very similar tropospheric bromine-chemistry scheme, it is likely that the major factor causing model-to-model differences is in the physical set-up, rather than the chemical set-up. This is reasonable given that p-TOMCAT is a CTM while UKCA is a GCM."

*Question: Page 7, line 32: Effect of dry deposition on ozone abundances during long-range transport into the Arctic. A key references here showing suppressed high latitude ozone due to deposition loss to vegetation in Siberia is Stjernberg et al., (2012).*

Answer: In the revision we added the following sentence: "For example, the suppressed high latitude summer ozone in Siberia is related to deposition loss to vegetation during long-range transport into the Arctic (Engvall Stjernberg et al., 2012)."

*Question: Section 4.1: Discussion of model and observed seasonal cycles. I think it is worth explicitly pointing out that inclusion of the halogen chemistry the control pTOMCAT model leads to severe underestimation of spring ozone at Summit and Pallas.*

Answer: Point taken. We clearly mentioned it in the revision.

*Question: It does not seem obvious to me why in presentation of the results in Section 4.3.1 model output switches from using p-TOMCAT to UKCA. Perhaps I have missed something in the applicability of the simulations to different periods. In general, only one model is shown for each part of the results comparing with observations. Would it be more informative to show both models where possible?*

Answer: In the revision, we showed both models outputs for all the cases that were discussed in the manuscript (see below).

[Figure]

New figure 8: Same as Fig.7 but for UKCA_control run result.

[Figure]

New figure 10: Same as Fig. 9 but for UKCA_control run result.

[Figure]

New Fig. 13: Same as Fig.12 but for pTOMCAT_control run result.

[Figure]

New Fig.15: Same as Fig. 14 but for pTOMCAT_control run result.

*Question: Throughout: I find the use of the labels "pTOMCAT_SI_OO_VSLS" and "UKCA_SI_OO_VSLS" to name the two control runs overly complicated and distracting when reading. It is better just to call these "pTOMCAT_control" and "UKCA_control" in the text. The names of the other experiments are then enough to highlight what is missing/included for the other runs.*

Answer: Point accepted, and we have used these new labels in the revision.

*Question: I would recommend a more explicit short "Summary" or "Conclusions" section to definitively set out the key findings of the study and their context in the wider picture. At the moment, the final paragraph is a bit brief and needs to be separated from the main discussion.*

Answer: Thank you for the constructive suggestion, in the revision we have re-written the 5: Discussion section and the 6: Summary section, shown below:

[revised manuscript text omitted]

*Question: Figure 2 caption: "in various experiments" – please clarify in the caption from which model.*

Answer: Done.

*Question: Figure 6 caption, please correct date "(May 201)"*

Answer: Done.

---

## Author Response (AR1)

**Reviewer's comments:** *These other mechanisms were cited, in the introduction, but the citation didn't really explain that snowpack mechanisms have been shown to activation halogens.*

**Answer:** In the revised version uploaded, we added one sentence in the introduction section (Page 2 Line: 28-29) "
[revised manuscript text omitted]